# Distributional Model Equivalence for Risk-Sensitive Reinforcement Learning

**Tyler Kastner**[*]
University of Toronto, Vector Institute

**Murat A. Erdogdu**
University of Toronto, Vector Institute

**Amir-massoud Farahmand**
Vector Institute, University of Toronto

## Abstract

We consider the problem of learning models for risk-sensitive reinforcement learning. We theoretically demonstrate that proper value equivalence, a method of learning models which can be used to plan optimally in the risk-neutral setting, is not sufficient to plan optimally in the risk-sensitive setting. We leverage distributional reinforcement learning to introduce two new notions of model equivalence, one which is general and can be used to plan for any risk measure, but is intractable; and a practical variation which allows one to choose which risk measures they may plan optimally for. We demonstrate how our framework can be used to augment any model-free risk-sensitive algorithm, and provide both tabular and large-scale experiments to demonstrate its ability.

## 1   Introduction

Reinforcement learning is a general framework where agents learn to sequentially make decisions to optimize an objective, such as the expected value of future rewards (risk-neutral objective) or the conditional value at risk of future rewards (risk-sensitive objective). It is a popular belief that a truly general agent must have a world model to plan with and limit the number of environment interactions needed (Russell, 2010). One way this is achieved is through model-based reinforcement learning, where an agent learns a model of the environment as well as its policy which it uses to act.

The classical approach to learning a model is to use maximum likelihood estimation (MLE) based on data, which given a model class selects the model which is most likely to have produced the data seen. If the model class is expressive enough, and there is enough data, we may expect a model learnt using MLE to be useful for risk-sensitive planning. However, the success of this method relies on the model being able to model everything about the environment, which is an unrealistic assumption in general.

As opposed to learning models which are accurate in modelling every aspect of the environment, recent works have advocated for learning models with the decision problem in mind, known as decision-aware model learning (Farahmand et al., 2017; Farahmand, 2018; D'Oro et al., 2020; Abachi et al., 2020; Grimm et al., 2020, 2021). In particular, Farahmand et al. (2017) introduced *value-aware model learning*, which uses a model loss that weighs model errors based on the effect the errors have on potential value functions. This framework has since been iterated on and improved upon in later works such as Farahmand (2018); Abachi et al. (2020); Voelcker et al. (2021). Complementarily, Grimm et al. (2020) introduced the *value equivalence principle*, a framework of partitioning the space of models based on the properties of the Bellman operators they induce. This framework has been extended by Grimm et al. (2021), where the authors introduce a related partitioning, called

---

[*]Correspondence to: tkastner@cs.toronto.edu.

37th Conference on Neural Information Processing Systems (NeurIPS 2023).

*proper value equivalence*, based on which models induce the same value functions. They substantiate this approach by demonstrating that any model in the same equivalence class as the true model is sufficient for optimal planning.

While standard reinforcement learning maximizes the expected return achieved by an agent, this may not suffice for many real-life applications. When environments are highly stochastic or where safety is important, a trade-off between the expected return and its variability is often desired. This concept is well-established in finance, and is the basis of modern portfolio theory (Markowitz, 1952). Recently this approach has been used in reinforcement learning, and is referred to as *risk-sensitive reinforcement learning*. In this setting, agents learn to maximise a *risk measure* of the return which is possibly different from expectation (in the case it is expectation, it is referred to as risk-neutral), and may penalize or reward risky behaviour (Howard & Matheson, 1972; Heger, 1994; Tamar et al., 2015, 2012; Chow et al., 2015; Tamar et al., 2016). In particular, Grimm et al. (2021, 2022) has explored when optimal risk-neutral planning in an approximate model translates to optimal behaviour in the true environment. However, it is not clear when this holds for risk-sensitive planning.

In this paper, we propose a framework that consolidates risk-sensitive reinforcement learning and decision-aware model learning. Specifically, we address the following question: *if we can perform risk-sensitive planning in an approximate model, does it translate to risk-sensitive behaviour in the true environment?* To this end, our work provides the following contributions:

- We prove that proper value equivalence only suffices for optimal planning in the risk-neutral case, and the performance of risk-sensitive planning decreases with risk-sensitivity (Section 3).
- We introduce the distribution equivalence principle, and show that this suffices for optimal planning with respect to *any* risk measure (Section 4).
- We introduce an approximate version of distribution equivalence, which is applicable in practice, that allows one to choose which risk measures they may plan optimally for (Section 5).
- We discuss how these methods may be learnt via losses, and how it can be combined with any existing model-free algorithm (Section 6).
- We demonstrate our framework empirically in both tabular and continuous domains (Section 7).

**Notation**

We write $\mathscr{P}(\mathcal{Z})$ to represent the set of probability measures on a measurable set $\mathcal{Z}$. For a probability measure $\nu \in \mathscr{P}(\mathcal{Z})$, we write $X \sim \nu$ to denote a random variable $X$ with law $\nu$, meaning for all measurable subsets $A \subseteq \mathcal{Z}$, $\mathbb{P}(X \in A) = \nu(A)$. For a probability measure $\nu \in \mathscr{P}(\mathcal{Z})$ and a measurable function $f : \mathcal{Z} \to \mathcal{Y}$, the pushforward measure $f_{\#}\nu \in \mathscr{P}(\mathcal{Y})$ is defined by $f_{\#}\nu(Y) = \nu(f^{-1}(Y))$ for all measurable sets $Y \subseteq \mathcal{Y}$. For arbitrary sets $X$ and $Y$, we write $Y^X$ for the space of functions from $X$ to $Y$.

## 2 Background

We consider a Markov decision process (MDP) represented as a tuple $(\mathcal{X}, \mathcal{A}, \mathcal{P}, \mathcal{R}, \gamma)$ where $\mathcal{X}$ is the state space, $\mathcal{A}$ is the action space, $\mathcal{P} : \mathcal{X} \times \mathcal{A} \to \mathscr{P}(\mathcal{X})$ is the transition kernel, $\mathcal{R} : \mathcal{X} \times \mathcal{A} \to \mathscr{P}(\mathbb{R})$ is the reward kernel, and $\gamma \in [0, 1)$ is the discount factor. We define a policy to be a map $\pi : \mathcal{X} \to \mathscr{P}(\mathcal{A})$, and write the set of all policies as $\Pi$. Given a policy $\pi \in \Pi$, we can sample trajectories $(X_t, A_t, R_t)_{t \geq 0}$, where for all $t \geq 0$, $A_t \sim \pi(\cdot \,|\, X_t)$, $R_t \sim \mathcal{R}(X_t, A_t)$, and $X_{t+1} \sim \mathcal{P}(X_t, A_t)$. For a trajectory from $\pi$ beginning at $X_0 = x$, we associate to it the return random variable $G^\pi(x) = \sum_{t \geq 0} \gamma^t R_t$. The expected return across all trajectories starting from a state $x$ is the value function $V^\pi(x) = \mathbb{E}_\pi[G^\pi(x)]$. The value function is the unique fixed point of the Bellman operator $T^\pi : \mathbb{R}^{\mathcal{X}} \to \mathbb{R}^{\mathcal{X}}$, defined by

$$T^\pi V(x) \triangleq \mathbb{E}_\pi \left[ R + \gamma V(X') \right], \tag{1}$$

where $\mathbb{E}_\pi$ is written to indicate $A \sim \pi(\cdot \,|\, x)$, $R \sim \mathcal{R}(x, A)$, and $X' \sim \mathcal{P}(x, A)$.

### 2.1 Model-based reinforcement learning and the value equivalence principle

Estimating (1) in the RL setting is not possible directly, as generally an agent does not have access to $\mathcal{R}$ nor $\mathcal{P}$, but only samples from them. There are two common approaches to address this: model-free

methods estimate the expectations through the use of stochastic approximation or related methods (Sutton, 1988), while model-based approaches learn an approximate model $\tilde{\mathcal{R}}, \tilde{\mathcal{P}}$ (Sutton, 1991).

We will refer to a tuple $\tilde{m} = (\tilde{\mathcal{R}}, \tilde{\mathcal{P}})$ as a model, and write $\mathbb{M}$ for the set of all models. In turn, each model $\tilde{m}$ induces an approximate MDP $(\mathcal{X}, \mathcal{A}, \tilde{\mathcal{P}}, \tilde{\mathcal{R}}, \gamma)$. For a policy $\pi$, we write $T_{\tilde{m}}^{\pi} : \mathbb{R}^{\mathcal{X}} \to \mathbb{R}^{\mathcal{X}}$ for the Bellman operator in this approximate MDP, and we write $V_{\tilde{m}}^{\pi}$ for the unique fixed point of this operator. We write $m^* = (\mathcal{R}, \mathcal{P})$ for the true model, and keep $T^{\pi} = T_{m^*}^{\pi}$. Throughout the paper, we will write $\mathcal{M} \subseteq \mathbb{M}$ to represent a set of models which we are considering.

Traditional methods of model-based reinforcement learning learn a model $\tilde{m}$ using task-agnostic methods such as maximum likelihood estimation (Sutton, 1991; Parr et al., 2008; Oh et al., 2015). More recent approaches have focused on learning models which are accurate in aspects which are necessary for decision making (Farahmand et al., 2017; Farahmand, 2018; Schrittwieser et al., 2020; Grimm et al., 2020, 2021; Arumugam & Van Roy, 2022). Of importance to us is Grimm et al. (2021), which introduced proper value equivalence, and defined the set $\mathcal{M}^{\infty}(\Pi) \triangleq \{\tilde{m} \in \mathcal{M} : V^{\pi} = V_{\tilde{m}}^{\pi}, \ \forall \pi \in \Pi\}$. They proved that any model $\tilde{m} \in \mathcal{M}^{\infty}(\Pi)$ suffices for optimal planning, that is, a policy which is optimal in $\tilde{m}$ is also optimal in the true environment, $m^*$.

## 2.2 Distributional reinforcement learning

Distributional reinforcement learning (Morimura et al., 2010; Bellemare et al., 2017, 2023) studies the return $G^{\pi}$ as a random variable, rather than focusing only on its expectation. For $x \in \mathcal{X}$, we define the return distribution $\eta^{\pi}(x)$ as the law of the random variable $G^{\pi}(x)$. The return distribution is the unique fixed point of the distributional Bellman operator $\mathcal{T}^{\pi} : \mathscr{P}(\mathbb{R})^{\mathcal{X}} \to \mathscr{P}(\mathbb{R})^{\mathcal{X}}$ given by

$$\mathcal{T}^{\pi}\eta(x) \triangleq \mathbb{E}_{\pi}\left[(b_{R,\gamma})_{\#}\eta(X')\right],$$

where $b_{R,\gamma} : x \mapsto R + \gamma x$ and $\mathbb{E}_{\pi}$ is as in (1). As was the case in Section 2.1, any approximate model $\tilde{m}$ induces a distributional Bellman operator $\mathcal{T}_{\tilde{m}}^{\pi}$, and we write $\eta_{\tilde{m}}^{\pi}$ for the unique fixed point of this operator.

## 2.3 Risk-sensitive reinforcement learning

We define a risk measure to be a function $\rho : \mathscr{P}_{\rho}(\mathbb{R}) \to [-\infty, \infty)$, where $\mathscr{P}_{\rho}(\mathbb{R}) \subseteq \mathscr{P}(\mathbb{R})$ is its domain.[2]. A classic example is $\rho = \mathbb{E}$, which we refer to as the *risk-neutral* case. When $\rho$ depends on more than only the mean of a distribution, we refer to $\rho$ as being *risk-sensitive*. The area of risk-sensitive reinforcement learning is concerned with maximizing various risk measures of the random return, rather than the expectation as done classically. We now present two examples of commonly used risk measures.

**Example 2.1.** For $\lambda > 0$, the mean-variance risk criterion is given by $\rho_{\mathrm{MV}}^{\lambda}(\mu) = \mathbb{E}_{Z \sim \mu}[Z] - \lambda \mathrm{Var}_{Z \sim \mu}(Z)$ (Markowitz, 1952; Tamar et al., 2012). This forms the basis of modern portfolio theory (Elton & Gruber, 1997).

**Example 2.2.** The conditional value at risk at level $\tau \in [0, 1]$ is defined as

$$\mathrm{CVaR}_{\tau}(\mu) \triangleq \frac{1}{\tau} \int_0^{\tau} F_{\mu}^{-1}(u) \, du,$$

where $F_{\mu}^{-1}(u) = \inf\{z \in \mathbb{R} : \mu(-\infty, z] \geq u\}$ is the quantile function of $\mu$. If $F_{\mu}^{-1}$ is a strictly increasing function, we equivalently have

$$\mathrm{CVaR}_{\tau}(\mu) = \mathbb{E}_{Z \sim \mu}\left[Z \,\middle|\, Z \leq F_{\mu}^{-1}(\tau)\right],$$

so that $\mathrm{CVaR}_{\tau}(\mu)$ can be understood as the expectation of the lowest $(100 \cdot \tau)\%$ of samples from $\mu$.

We say that a policy $\pi_{\rho}^*$ is optimal with respect to $\rho$ if

$$\rho\left(\eta^{\pi_{\rho}^*}(x)\right) = \max_{\pi \in \Pi} \rho\left(\eta^{\pi}(x)\right), \forall x \in \mathcal{X}.$$

---

[2]We use the definition of risk measure used by Bellemare et al. (2023). In earlier financial mathematics literature such as Artzner et al. (1999), risk measures were defined as functions of random variables, rather than probability measures. By defining the domain to be a subset of probability measures, we are implicitly considering law-invariant risk measures (Kusuoka, 2001).

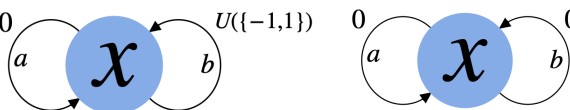

Figure 1: An MDP with a single state and two actions (left), and a proper value equivalent model $\tilde{m}$ for it (right).

Since we define the space of policies as $\Pi = \mathscr{P}(\mathcal{A})^{\mathcal{X}}$, we implicitly only considering the class of stationary Markov policies (Puterman, 2014). For a general risk measure, an optimal policy in this class may not exist (Bellemare et al., 2023). We discuss more general policies in Appendix D.

## 3    Limitations of value equivalence for risk-sensitive planning

Grimm et al. (2021) proved that any proper value equivalent model is sufficient for optimal risk-neutral planning. In this section, we investigate whether this holds for risk-sensitive planning as well, or is limited to the risk-neutral setting.

As an illustrative example, let us consider the MDP and approximate model $\tilde{m}$ in Figure 1. It is straightforward to verify that $\tilde{m}$ is a proper value equivalent model for the true MDP, as the value for any policy is 0 in both $\tilde{m}$ and the true environment. However, for a risk-sensitive agent $\tilde{m}$ is not sufficient: the variability of return when choosing action $b$ in $m^*$ is much higher than the variability of return when choosing action $b$ in $\tilde{m}$. Formally, let us fix $\gamma = \frac{1}{2}$, and let $\pi^b$ be the policy which chooses action $b$ with probability 1. Then $\eta^{\pi^b}(x) = U([-2, 2])$ (Bellemare et al., 2023, Example 2.10), while $\eta_{\tilde{m}}^{\pi^b}(x) = \delta_0$ (where $\delta_x$ refers to the Dirac distribution concentrated at $x$). This difference prevents $\tilde{m}$ from planning optimally for risk-sensitive risk measures. For example, the optimal policy with respect to $\rho_{\mathrm{MV}}^{\lambda}$ in $m^*$ is to choose $a$ with probability 1, while in $\tilde{m}$ any policy is optimal. It is straightforward to validate that similar phenomena happen for $\mathrm{CVaR}_\tau$ when $\tau < 1$.

As demonstrated in the example above, proper value equivalence is not sufficient for planning with respect to the risk measures introduced in Section 2.3. We now formalize this, and demonstrate that the only risk measures which proper value equivalence can plan for exactly are those which are functions of expectation.

**Proposition 3.1.** *Let $\rho$ be a risk measure such that for any MDP and any set $\mathcal{M}$ of models, a policy optimal for $\rho$ for any $\tilde{m} \in \mathcal{M}^\infty(\Pi)$ is optimal in the true MDP. Then $\rho$ must be risk-neutral, in the sense that there exists an increasing function $g : \mathbb{R} \to \mathbb{R}$ such that $\rho(\nu) = g(\mathbb{E}_{Z \sim \nu}[Z])$.*

The previous proposition demonstrates that in general, the only risk measures we can plan for in a proper value equivalent model are those which are transformations of the value function. However, it does not address the question of how well proper value equivalent models can be used to plan with respect to other risk measures.

To investigate this question, we turn our attention to a class of risk measures known as *spectral risk measures* (Acerbi, 2002). Let $\varphi : [0, 1] \to \mathbb{R}$ be a non-negative, non-increasing, right-continuous, integrable function such that $\int_0^1 \varphi(u)\, du = 1$. Then the spectral risk measure corresponding to $\varphi$ is defined as

$$\rho_\varphi(\nu) \triangleq \int_0^1 F_\nu^{-1}(u)\varphi(u)du,$$

where $F_\nu^{-1}$ is as in Example 2.2. Spectral risk measures encompass many common risk measures, for example choosing $\varphi = \mathbb{1}_{[0,1]}$ corresponds to expectation, while $\varphi = \frac{1}{\tau}\mathbb{1}_{[0,\tau]}$ corresponds to $\mathrm{CVaR}_\tau$.

We say that a spectral risk measure $\rho$ is $(\varepsilon, \delta)$-strictly risk-sensitive if it corresponds to a function $\varphi$ such that $\varphi(\varepsilon) \leq \delta$. This implies that $\varphi$ gives a weight of less than $\delta$ to the top $\varepsilon$ quantiles, and so intuitively a larger $\varepsilon$ and smaller $\delta$ implies a more risk-sensitive risk measure.

With this definition, we now demonstrate that when using proper value equivalent models to plan for strictly risk-sensitive spectral risk measures, there exists a tradeoff between the level of risk-sensitivity and the performance achieved.

**Proposition 3.2.** *Let $\rho$ be an $(\varepsilon, \delta)$-strictly risk-sensitive spectral risk measure, and suppose that rewards are almost surely bounded by $R_{max}$. Then there exists an MDP with a proper value equivalent model $\tilde{m}$ with the following property: letting $\pi_\rho^*$ be an optimal policy for $\rho$ in the original MDP, and $\tilde{\pi}_\rho^*$ an optimal policy for $\rho$ in $\tilde{m}$, we have*

$$\inf_{x \in \mathcal{X}} \left\{ \rho\big(\eta^{\pi_\rho^*}(x)\big) - \rho\big(\eta^{\tilde{\pi}_\rho^*}(x)\big) \right\} \geq \frac{R_{max}}{1 - \gamma} \varepsilon(1 - \delta(1 - \varepsilon)).$$

The fact that we take an infimum over $\mathcal{X}$ is important to note: there exists an MDP such that for *any* state $x$, the performance gap due to planning in the proper value equivalent model is at least $\frac{R_{\max}}{1-\gamma} \varepsilon(1 - \delta(1 - \varepsilon))$.. This weakness motivates us to introduce a new notion of model equivalence.

# 4 The distribution equivalence principle

We now introduce a novel notion of equivalence on the space of models, which can be used for risk-sensitive learning. Intuitively, proper value equivalence ensures matching of the *means* of the approximate and true return distributions, which is why it can only produce optimal policies for risk measures which depend on the mean. In order to plan for any risk measure, we leverage the distributional perspective of RL, to partition models based on their *entire* return distribution.

**Definition 4.1.** Let $\Pi \subseteq \mathbb{\Pi}$ be a set of policies and $\mathcal{D} \subseteq \mathscr{P}(\mathbb{R})^{\mathcal{X}}$ be a set of distribution functions. We say that the space of *distribution equivalent* models with respect to $\Pi$ and $\mathcal{D}$ is

$$\mathcal{M}_{\text{dist}}(\Pi, \mathcal{D}) \triangleq \{\tilde{m} \in \mathcal{M} : \mathcal{T}^\pi \eta = \mathcal{T}_{\tilde{m}}^\pi \eta, \ \forall \pi \in \Pi, \eta \in \mathcal{D}\}.$$

We can extend this concept to equivalence over multiple applications of the Bellman operator. Following this, for $k \in \mathbb{N}$ we define the order $k$ distribution-equivalence class as

$$\mathcal{M}_{\text{dist}}^k(\Pi, \mathcal{D}) \triangleq \left\{\tilde{m} \in \mathcal{M} : (\mathcal{T}^\pi)^k \eta = (\mathcal{T}_{\tilde{m}}^\pi)^k \eta, \ \forall \pi \in \Pi, \forall \eta \in \mathcal{D}\right\}.$$

Taking the limit as $k \to \infty$, we retrieve the set of proper distribution equivalent models.

**Definition 4.2.** Let $\Pi \subseteq \mathbb{\Pi}$ be a set of policies. We define the set of *proper distribution equivalent* models with respect to $\Pi$ as

$$\mathcal{M}_{\text{dist}}^\infty(\Pi) \triangleq \{\tilde{m} \in \mathcal{M} : \eta_{\tilde{m}}^\pi = \eta^\pi, \ \forall \pi \in \Pi\}.$$

As discussed in Section 3, models in $\mathcal{M}^\infty(\mathbb{\Pi})$ are sufficient for optimal planning with respect to expectation, but generally not with respect to other risk measures. We now show that proper distribution equivalence removes this problem: choosing a model in $\mathcal{M}_{\text{dist}}^\infty(\mathbb{\Pi})$ is sufficient for optimal planning with respect to *any* risk measure.

**Theorem 4.3.** *Let $\rho$ be any risk measure. Then an optimal policy with respect to $\rho$ in $\tilde{m} \in \mathcal{M}_{\text{dist}}^\infty(\mathbb{\Pi})$ is optimal with respect to $\rho$ in $m^*$.*

At this point, it appears that distribution equivalence addresses nearly all of the limitations of value equivalence discussed in Section 3. However, the nature of distributions brings inherent challenges, in particular they are infinite dimensional. As a result of this, for computational purposes one must use a parametric family of distributions $\mathscr{F} \subseteq \mathscr{P}(\mathbb{R})$ (Rowland et al., 2018; Dabney et al., 2018) to represent return distributions. However, an additional challenge is that the distributional Bellman operator may bring return distributions out of the parametric representation space: for a general $\eta \in \mathscr{F}^{\mathcal{X}}, \mathcal{T}^\pi \eta \notin \mathscr{F}^{\mathcal{X}}$. Hence, we also require a projection operator[3] $\Pi_{\mathscr{F}} : \mathscr{P}(\mathbb{R})^{\mathcal{X}} \to \mathscr{F}^{\mathcal{X}}$, and in practice we must use $\Pi_{\mathscr{F}} \mathcal{T}^\pi \eta$. This also implies that it may not be feasible to learn a model $\tilde{m}$ in $\mathcal{M}_{\text{dist}}^k(\Pi, \mathcal{D})$ or $\mathcal{M}_{\text{dist}}^\infty(\Pi)$: they rely on matching $\mathcal{T}^\pi \eta$ or $\eta^\pi$, while one would only have access to $\Pi_{\mathscr{F}} \mathcal{T}^\pi \eta$ and $\Pi_{\mathscr{F}} \eta^\pi$. We address this issue next, through the perspective of *statistical functionals*.

# 5 Statistical functional equivalence

Following the intractability of learning a distribution equivalent model in practice, we now study model equivalence through the lens of *statistical functionals*, a framework introduced by Rowland

---

[3]Further details on the necessity of the projection operator and a discussion of various projections can be found in Chapter 5 of Bellemare et al. (2023).

et al. (2019) to describe a variety of distributional reinforcement learning algorithms. We begin with a review of statistical functionals (Section 5.1), and then introduce statistical functional equivalence, demonstrate its equivalence to projected distribution equivalence, and study which risk measures it can plan optimally for (Section 5.2).

## 5.1 Background on statistical functionals

**Definition 5.1.** A statistical functional is a function $\psi : \mathscr{P}_\psi(\mathbb{R}) \to \mathbb{R}$, where $\mathscr{P}_\psi(\mathbb{R}) \subseteq \mathscr{P}(\mathbb{R})$ is its domain. A sketch is a collection of statistical functionals, written as a mapping $\psi : \mathscr{P}_\psi(\mathbb{R}) \to \mathbb{R}^m$, where $\psi = (\psi_1, \ldots, \psi_m)$, and $\mathscr{P}_\psi(\mathbb{R}) = \bigcap_{i=1}^m \mathscr{P}_{\psi_i}(\mathbb{R})$.

**Example 5.2.** Suppose $i > 0$, and let $\mathscr{P}_i(\mathbb{R})$ be the set of probability measures with finite $i$th moment. Moreover, let $\mu_i(\nu)$ be the $i$th moment of a measure $\nu \in \mathscr{P}_i(\mathbb{R})$. Then for $m > 0$, the $m$ moment sketch $\psi_\mu^m : \mathscr{P}_m(\mathbb{R}) \to \mathbb{R}^m$ is defined by $\psi_\mu^m(\nu) = (\mu_1(\nu), \ldots, \mu_m(\nu))$.

For a given sketch $\psi$, we define its image as $I_\psi = \{\psi(\nu) : \nu \in \mathscr{P}_\psi(\mathbb{R})\} \subseteq \mathbb{R}^m$. An imputation strategy for a sketch $\psi$ is a map $\iota : I_\psi \to \mathscr{P}_\psi(\mathbb{R})$, and can be thought of as an approximate inverse (a true inverse may not exist as $\psi$ is generally not injective). We say $\iota$ is *exact* for $\psi$ if for any $(s_1, \ldots, s_m) \in I_\psi$ we have $(s_1, \ldots, s_m) = \psi(\iota(s_1, \ldots, s_m))$. In general, an exact imputation strategy always exists, however it may not be efficiently computable (Bellemare et al., 2023).

**Example 5.3.** Suppose $\psi$ is a sketch given by $\psi(\nu) = (\mathbb{E}_{Z \sim \nu}[Z], \mathrm{Var}_{Z \sim \nu}[Z])$, and $\iota$ is given by $\iota(\mu, \sigma^2) = \mathcal{N}(\mu, \sigma^2)$ (that is, the normal distribution with mean $\mu$ and variance $\sigma^2$). One may verify that $\iota$ is exact, since for any $(\mu, \sigma^2) \in \mathbb{R}^2 = I_\psi$, we have $\psi(\iota(\mu, \sigma^2)) = (\mu, \sigma^2)$.

We now extend the notion of statistical functionals to return-distribution functions. For $\eta \in \mathscr{P}_\psi(\mathbb{R})^{\mathcal{X}}$ we write $\psi(\eta) = (\psi(\eta(x)) : x \in \mathcal{X})$. We say that a set $\Omega \subseteq \mathscr{P}(\mathbb{R})$ is closed under $\mathcal{T}^\pi$ if whenever $\eta \in \Omega^{\mathcal{X}}$, we have $\mathcal{T}^\pi \eta \in \Omega^{\mathcal{X}}$. A sketch $\psi$ is Bellman-closed (Rowland et al., 2019; Bellemare et al., 2023) if whenever its domain is closed under $\mathcal{T}^\pi$, there exists an operator $\mathcal{T}_\psi^\pi : I_\psi^{\mathcal{X}} \to I_\psi^{\mathcal{X}}$ such that for any $\eta \in \mathscr{P}_\psi(\mathbb{R})^{\mathcal{X}}$,

$$\psi(\mathcal{T}^\pi \eta) = \mathcal{T}_\psi^\pi \psi(\eta).$$

We refer to $\mathcal{T}_\psi^\pi$ as the Bellman operator for $\psi$. Similarly to Section 2.1, we denote $\mathcal{T}_{\psi,\tilde{m}}^\pi$ as the Bellman operator for $\psi$ in an approximate model $\tilde{m}$.

We will write $s_\psi^\pi = \psi(\eta^\pi)$ as a shorthand, and refer to it as the *return statistic* for a policy $\pi$. If $\mathcal{T}_\psi^\pi$ exists, then $s_\psi^\pi$ is its fixed point: $s_\psi^\pi = \mathcal{T}_\psi^\pi s_\psi^\pi$. For an approximate model $\tilde{m}$, we write $s_{\psi,\tilde{m}}^\pi = \psi(\eta_{\tilde{m}}^\pi)$. We further have $s_{\psi,\tilde{m}}^\pi = \mathcal{T}_{\psi,\tilde{m}}^\pi s_{\psi,\tilde{m}}^\pi$, that is, it is a fixed point of the Bellman operator $\mathcal{T}_{\psi,\tilde{m}}^\pi$.

The task of policy evaluation for a statistical functional $\psi$ is that of computing the value $s_\psi^\pi$. Statistical functional dynamic programming (Bellemare et al., 2023) aims to do this by computing the iterates $s_{k+1} = \psi(\mathcal{T}^\pi \iota(s_k))$, with $s_0 \in I_\psi^{\mathcal{X}}$ initialized arbitrarily. If $\iota$ is exact and $\psi$ is Bellman-closed, then the updates satisfy $s_k = \psi(\eta_k)$, where $\eta_0 = \iota(s_0)$ and $\eta_{k+1} = \mathcal{T}^\pi \eta_k$. If $\psi$ is a continuous sketch[4], then the iterates $(s_k)_{k \geq 0}$ converge to $s_\psi^\pi$.

## 5.2 Statistical functional equivalence

We now introduce a notion of model equivalence through the lens of statistical functionals. Intuitively, this allows us to interpolate between value equivalence and distribution equivalence, as we can choose exactly which aspects of the return distributions we would like to capture.

**Definition 5.4.** Let $\psi$ be a sketch, and $\iota$ be an imputation strategy for $\psi$. Let $\mathcal{I} \subseteq I_\psi^{\mathcal{X}}$ and $\Pi \subseteq \mathbb{\Pi}$. We define the class of $\psi$ *equivalent models* with respect to $\Pi$ and $\mathcal{I}$ as

$$\mathcal{M}_\psi(\Pi, \mathcal{I}) \triangleq \left\{ \tilde{m} \in \mathcal{M} : \psi\left(\mathcal{T}^\pi \iota(s)\right) = \psi\left(\mathcal{T}_{\tilde{m}}^\pi \iota(s)\right), \forall \pi \in \Pi, \forall s \in \mathcal{I} \right\}.$$

In the case that $\psi$ is Bellman-closed and $\iota$ is exact, this set can be described in a form similar to that of value equivalence and distribution equivalence.

---

[4]We define this notion in Appendix A.2.

**Proposition 5.5.** *If $\psi$ is Bellman-closed and $\iota$ is exact, we have that*

$$\mathcal{M}_\psi(\Pi, \mathcal{I}) = \left\{ \tilde{m} \in \mathcal{M} : \mathcal{T}_\psi^\pi s = \mathcal{T}_{\psi, \tilde{m}}^\pi s, \ \forall \pi \in \Pi, \forall s \in \mathcal{I} \right\}.$$

We can extend the above to $k$ applications of the projected Bellman operator, and define the set of order-$k$ $\psi$ equivalent models as

$$\mathcal{M}_\psi^k(\Pi, \mathcal{I}) \triangleq \left\{ \tilde{m} \in \mathcal{M} : (\psi\mathcal{T}^\pi\iota)^k\, s = (\psi\mathcal{T}_{\tilde{m}}^\pi\iota)^k\, s, \forall \pi \in \Pi, \forall s \in \mathcal{I} \right\},$$

where $\psi\mathcal{T}^\pi\iota : I_\psi^\mathcal{X} \to I_\psi^\mathcal{X}$ is shorthand for $s \mapsto \psi(\mathcal{T}^\pi\iota(s))$. As in Proposition 5.5, if $\psi$ is Bellman-closed and $\iota$ is exact, it holds that

$$\mathcal{M}_\psi^k(\Pi, \mathcal{I}) = \left\{ \tilde{m} \in \mathcal{M} : (\mathcal{T}_\psi^\pi)^k s = (\mathcal{T}_{\psi, \tilde{m}}^\pi)^k s, \ \forall \pi \in \Pi, \forall s \in \mathcal{I} \right\}.$$

Following Section 4, we can consider the set of models which agree on *return statistics*, and have no dependence on the set $\mathcal{I}$. However, one difference in the case of statistical functionals is that it is not true in general that this is equal to the limit of $\mathcal{M}_\psi^k(\Pi, \mathcal{I})$. Intuitively, this is for the same reason that the iterates $(s_k)_{k \geq 0}$ of statistical functional dynamic programming do not always converge to $s_\psi^\pi$ (Section 5.1). We first introduce the definition of proper statistical functional equivalence, and then demonstrate when it is the limiting set in Proposition 5.7.

**Definition 5.6.** Let $\Pi \subseteq \mathbb{\Pi}$ be a set of policies, and $\psi$ be a sketch. We define the class of proper statistical functional equivalent models with respect to $\psi$ and $\Pi$ as

$$\mathcal{M}_\psi^\infty(\Pi) \triangleq \left\{ \tilde{m} \in \mathcal{M} : s_{\psi, \tilde{m}}^\pi = s_\psi^\pi, \ \forall \pi \in \Pi \right\}.$$

**Proposition 5.7.** *If $\psi$ is both continuous and Bellman-closed and $\iota$ is exact, then*[5]

$$\lim_{k \to \infty} \mathcal{M}_\psi^k(\Pi, \mathcal{I}) = \mathcal{M}_\psi^\infty(\Pi), \text{ for any } \mathcal{I} \subseteq I_\psi^\mathcal{X}.$$

*Remark* 5.8. Value equivalence (Grimm et al., 2020, 2021) can be seen as a special case of statistical functional equivalence, in the sense that if we choose $\psi = \mathbb{E}$, then we have $\mathcal{M}_\psi^k(\Pi) = \mathcal{M}^k(\Pi)$, for any $\Pi \subseteq \mathbb{\Pi}$ and $k \in [1, \infty]$.

### Connection to projected distribution equivalence

In Section 4, we remarked that distribution equivalence was difficult to achieve in practice, due to the fact that the space $\mathscr{P}(\mathbb{R})^\mathcal{X}$ was infinite dimensional, and we generally rely on a parametric family $\mathscr{F}$. We now demonstrate that the statistical functional perspective provides us a way to address this.

Let $\psi$ be a sketch and $\iota$ an imputation strategy. These induce the implied representation (Bellemare et al., 2023) given by $\mathscr{F}_\psi = \{\iota(s) : s \in I_\psi\}$, and the projection operator $\Pi_{\mathscr{F}_\psi} : \mathscr{P}_\psi(\mathbb{R}) \to \mathscr{F}_\psi$ given by $\Pi_{\mathscr{F}_\psi} = \iota \circ \psi$. We now show that through this construction, we can relate statistical functional model learning to projected distributional model learning with the projection $\Pi_{\mathscr{F}_\psi}$.

**Proposition 5.9.** *Suppose $\iota$ is injective, $\Pi \subseteq \mathbb{\Pi}$, $\mathcal{I} \subseteq I_\psi^\mathcal{X}$, and let $\mathcal{D}_\mathcal{I} = \{\iota(s) : s \in \mathcal{I}\} \subseteq \mathscr{P}_\psi(\mathbb{R})^\mathcal{X}$. Then*

$$\mathcal{M}_\psi(\Pi, \mathcal{I}) = \left\{ \tilde{m} \in \mathcal{M} : \Pi_{\mathscr{F}_\psi}\mathcal{T}^\pi\eta = \Pi_{\mathscr{F}_\psi}\mathcal{T}_{\tilde{m}}^\pi\eta, \forall \pi \in \Pi, \forall \eta \in \mathcal{D}_\mathcal{I} \right\},$$

*and*

$$\mathcal{M}_\psi^\infty(\Pi) = \left\{ \tilde{m} \in \mathcal{M} : \Pi_{\mathscr{F}_\psi}\eta^\pi = \Pi_{\mathscr{F}_\psi}\eta_{\tilde{m}}^\pi, \ \forall \pi \in \Pi \right\}.$$

### Risk-sensitive learning

We now study which risk measures we can plan optimally for using a model in $\mathcal{M}_\psi^\infty(\mathbb{\Pi})$. Intuitively, we will not be able to plan optimally for all risk measures (as was the case in Theorem 4.3), since this set only requires models to match the aspects of the return distribution captured by $\psi$. Indeed, we now show that the choice of $\psi$ exactly determines which risk measures can be planned for.

**Proposition 5.10.** *Let $\rho$ be a risk measure and let $\psi = (\psi_1, \ldots, \psi_m)$ be a sketch, and suppose that $\rho$ is in the span of $\psi$, in the sense that there exists $\alpha_0, \ldots \alpha_m \in \mathbb{R}$ such that for all $\nu \in \mathscr{P}_\psi(\mathbb{R}) \cap \mathscr{P}_\rho(\mathbb{R})$, $\rho(\nu) = \sum_{i=1}^m \alpha_i \psi_i(\nu) + \alpha_0$. Then any optimal policy with respect to $\rho$ in $\tilde{m} \in \mathcal{M}_\psi^\infty(\mathbb{\Pi})$ is optimal with respect to $\rho$ in $m^*$.*

---

[5]This is a set theoretic limit, and we review its definition in Definition C.1. Further details can be found in many texts on analysis or probability, for example Resnick (1999).

# 6 Learning statistical functional equivalent models

We now analyze how we may learn models in these classes in practice. As we have introduced a number of concepts and spaces of models, we only discuss here the spaces of models that are used in the empirical evaluation which follow, and we discuss the remainder of the spaces in Appendix B.

We focus on the case of learning a proper $\psi$-equivalent model. Such a model must satisfy $s_\psi^\pi = (\psi \mathcal{T}_{\tilde{m}}^\pi \iota)^k s_\psi^\pi$ for any policy $\pi$ (Proposition B.1), so that we can construct a loss by measuring the amount that this equality is violated by. However, the size of $\mathbb{\Pi}$ is exponential in $|\mathcal{X}|$, so we can approximate this by only measuring the amount of violation over a subset of policies $\Pi \subseteq \mathbb{\Pi}$. We can now formalize this concept as a loss.

**Definition 6.1.** Let $\psi$ be a sketch, and $\iota$ an imputation strategy. We define the loss for learning a proper-$\psi$ equivalent model as

$$\mathcal{L}_{\psi,\Pi,\infty}^k(\tilde{m}) \triangleq \sum_{\pi \in \Pi} \left\| s_\psi^\pi - (\psi \mathcal{T}_{\tilde{m}}^\pi \iota)^k s_\psi^\pi \right\|_2^2.$$

If $\psi$ is Bellman-closed this can be written without the need for $\iota$, by replacing $\psi \mathcal{T}_{\tilde{m}}^\pi \iota$ with $\mathcal{T}_{\psi,\tilde{m}}^\pi$.

This loss is amenable to tabular environments, as it requires knowledge of $s_\psi^\pi$, which can be learnt approximately using statistical functional dynamic programming. Despite this, the above approach can be further adapted to the deep RL setting, which we now discuss, and describe how our approach can be combined with existing model-free risk-sensitive algorithms.

We will assume the existence of a model-free risk-sensitive algorithm which satisfies the following properties: (i) it learns a policy $\pi$ using a replay buffer $\mathcal{D}$, and (ii) it learns an approximate statistical functional function $s_{\psi,\omega}^\pi$ (for example, any algorithm based upon C51 (Bellemare et al., 2017) or QR-DQN (Dabney et al., 2018) satisfies these assumptions), where we write $\omega$ to refer to the set of parameters it depends on, and to emphasize its difference with the true return statistic $s_\psi^\pi$. We will introduce a loss which learns an approximate model $\tilde{m}$, which can then be combined with the replay buffer $\mathcal{D}$ to use both experienced transitions and modelled transitions to learn $\pi$, as was done by e.g. Sutton (1991) or Janner et al. (2019). Following this, for a learnt model $\tilde{m}$ we introduce the approximate loss

$$\mathcal{L}_{\mathcal{D},\psi,\omega}(\tilde{m}) = \mathop{\mathbb{E}}_{\substack{(x,a,r,x') \sim \mathcal{D} \\ \tilde{x}' \sim \tilde{m}(\cdot|x,a)}} \left[ (s_{\psi,\omega}^\pi(x') - s_{\psi,\omega}^\pi(\tilde{x}'))^2 \right].$$

# 7 Empirical evaluation

We now empirically study our framework, and examine the phenomena discussed in the previous sections. We focus on two sets of experiments: the first is in tabular settings where we use dynamic programming methods to perform an analysis without the noise of gradient-based learning. The second builds upon Lim & Malik (2022), where we augment their model-free algorithm with our framework, and evaluate it on an option trading environment. We discuss training and environments details in Appendix E, and provide additional results in Appendix A.3. We provide the code used to run our experiments at github.com/tylerkastner/distribution-equivalence.

## 7.1 Experimental details

**Tabular experiments**

For each environment, we learn an MLE model, a proper value equivalent model using the method introduced in Grimm et al. (2021), and a $\psi_\mu^2$ equivalent model using $\mathcal{L}_{\psi_\mu^2,\Pi,\infty}^k$, where $\psi_\mu^m$ is the first $m$ moment functional (cf. Example 5.2), and $\Pi$ is a set of 1000 randomly sampled policies. For each model, we performed CVaR value iteration (Bellemare et al., 2023), and further performed CVaR value iteration in the true model, to produce three policies. We repeat the learning of the models across 20 independent seeds, and report the performance of the policies in Figure 2 (Left).

**Option trading**

Lim & Malik (2022) introduced a modification of QR-DQN which attempts to learn CVaR optimal policies, that they evaluate on an option trading environment (Chow & Ghavamzadeh, 2014; Tamar

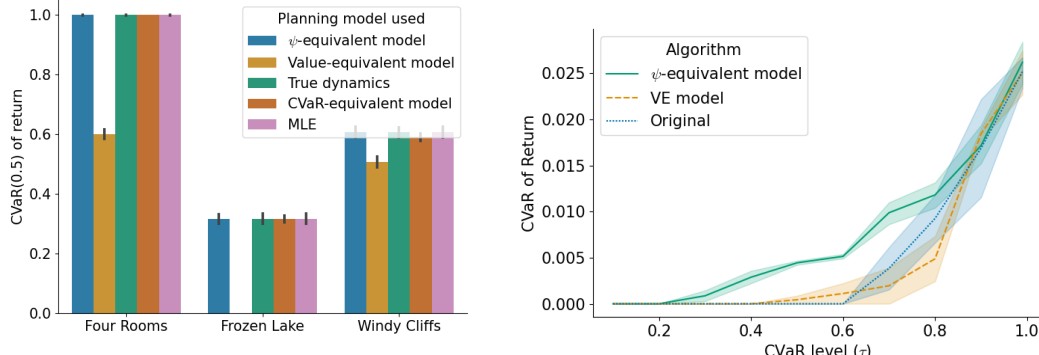

Figure 2: **Left**: CVaR(0.5) of returns obtained across the three tabular environments. We computed the values across 1000 trajectories from each of the 20 learnt models. Error bars indicate 95% confidence intervals. The orange bar for Frozen Lake appears missing because the value obtained is 0. **Right**: CVaR of returns for the policies learnt in the option trading environment for various CVaR levels after 10,000 environment interactions. Shaded regions indicate 95% confidence intervals across 10 independent seeds.

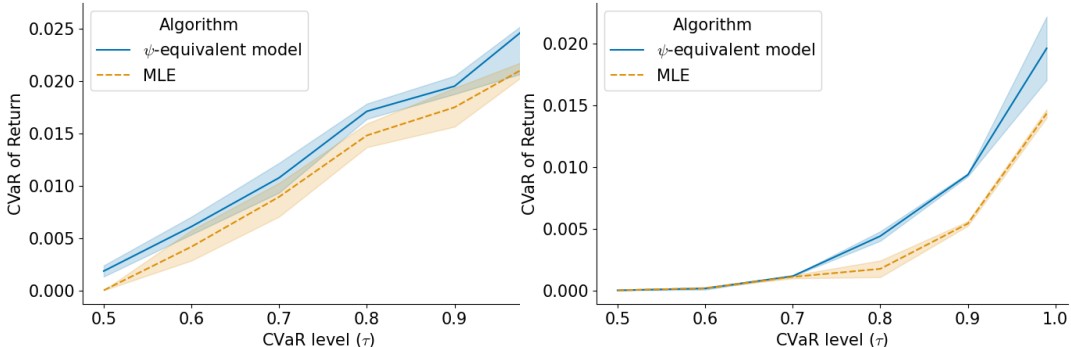

Figure 3: CVaR of returns for policies learnt in the option trading environment using a $\psi$-equivalent model and a MLE-based model as the number of distracting dimensions increases (**left**: 2 distracting dimensions, **right**: 6 distracting dimensions).

et al., 2017). We augment their method using the method described in Section 6, and we learn optimal policies for 10 CVaR levels between 0 and 1. We compare our adapted method to their original method as well as their original method adapted with a PVE model (Grimm et al., 2021), and discuss implementation details in Appendix E. In particular, we evaluate the models in a low-sample regime, so the sample efficiency gains of using a model are apparent.

## 7.2 Discussion

In Figure 2 (Left), we can see that across all three tabular environments, planning in a proper statistical functional equivalent model achieves stronger results over planning in a proper value equivalent model. This provides an empirical demonstration of Proposition 3.2 and Proposition 5.10: proper value equivalence is limited in its ability to plan risk-sensitively, while risk-sensitive planning in a statistical functional equivalent model approximates risk-sensitive planning in the true environment.

In Figure 2 (Right), we can see that Lim & Malik (2022)'s algorithm augmented with a statistical functional equivalent model achieved significantly improved performance for all CVaR levels below $\tau \approx 0.8$. The fact that our augmentation improves upon the original method reflects the improved sample efficiency which comes from using an approximate model for planning. This difference

is more apparent for lower values of $\tau$, which demonstrates the phenomenon that learning more risk-sensitive policies are less sample efficient (Greenberg et al., 2022). On the other hand, the method augmented with the PVE model could have the same sample efficiency gains from using an approximate model (as it is trained on the same number of environment interactions), so the fact that it is not performant for lower values of CVaR is a demonstration of Proposition 3.2: the more risk-sensitive the risk measure being planned for, the more the performance is affected.

Compared to MLE-based methods, our approach focuses on learning the aspects of the environment which are most relevant for risk-sensitive planning. This phenomenon is reflected in Figure 3, where the performance gap between the MLE model and the $\psi$-equivalent model grows with the amount of uninformative complexity in the environment.

## 8    Conclusion

In this work, we studied the intersection of model-based reinforcement learning and risk-sensitive reinforcement learning. We demonstrated that value-equivalent approaches to model learning produce policies which can only plan optimally for the risk-neutral setting, and in risk-sensitive settings their performance degrades with the level of risk being planned for. Similarly, we argued that MLE-based methods are insufficient for efficient risk-sensitive learning as they focus on all aspects of the environment equally. We then introduced distributional model equivalence, and demonstrated that distribution equivalent models can be used to plan for any risk measure, however they are intractable to learn in practice. To account for this, we introduced statistical functional equivalence; an equivalence which is parameterized by the choice of a statistical functional. We proved that the choice of statistical functional exactly determines which risk measures can be planned for optimally, and provided a loss with which these models can be learnt. We further described how our method can be combined with any existing model-free risk-sensitive algorithm, and augmented a recent model-free distributional risk-sensitive algorithm with our model. We supported our theory with empirical results, which demonstrated our approach's advantages over value-equivalence and MLE based models over a range of environments..

## 9    Acknowledgements

We would like to thank the members of the Adaptive Agents Lab who provided feedback on a draft of this paper. AMF acknowledges the funding from the Canada CIFAR AI Chairs program, as well as the support of the Natural Sciences and Engineering Research Council of Canada (NSERC) through the Discovery Grant program (2021-03701). MAE was supported by NSERC Grant [2019-06167], CIFAR AI Chairs program, and CIFAR AI Catalyst grant. Resources used in preparing this research were provided, in part, by the Province of Ontario, the Government of Canada through CIFAR, and companies sponsoring the Vector Institute. Finally, we thank the anonymous reviewers (both ICML 2023 and NeurIPS 2023) for providing feedback which improved the paper.

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

# A   Additional results

## A.1   Additional properties of statistical functional equivalence

We begin by discussion some additional properties of statistical functional equivalence.

**Proposition A.1.** *If the reward from a state is almost surely bounded, the $m$-moment functional $\psi_\mu^m$ is continuous on return distributions.*

*Proof.* Let $R_{\max}$ be the maximum absolute reward from a state, equivalently let us suppose the support of $\mathcal{R}(\cdot, \cdot)$ is a subset of $[-R_{\max}, R_{\max}]$. Then for any $x \in \mathcal{X}$ and any policy $\pi$ we have that the support of $\eta^\pi(x)$ is a subset of $[-R_{\max}/(1 - \gamma), R_{\max}/(1 - \gamma)]$.

We now leverage a result of Billingsley (1986), which states that for any $m > 0$, if a sequence of measures $(\nu_n)_{n \geq 0} \subseteq \mathscr{P}_m(\mathbb{R})$ is uniformly integrable, then $\mu_m(\nu_n)$ converges to $\mu_m(\nu)$ in $\mathbb{R}$ whenever $(\nu_n)_{n \geq 0}$ weakly converges to $\nu$. Applying this in our setting, we first fix $x \in \mathcal{X}$, then by the previous paragraph we have that the support of each $\eta_n(x)$ is a subset of the interval $[-R_{\max}/(1 - \gamma), R_{\max}/(1 - \gamma)]$, which implies uniform integrability of $(\eta_n(x))_{n \geq 0}$. Hence we have that $\mu_m(\eta_n(x)) \to \mu_m(\eta)$, which gives convergence of $\psi_\mu^m$. $\qquad\square$

**Example A.2.** Let $\psi_\mu^m$ be the sketch of the first $m$ moments (Example 5.2). Then by Proposition 5.10, whenever $m \geq 2$ we have that any proper $\psi_\mu^m$ equivalent model is sufficient for optimal planning with respect to the mean-variance risk criterion $\rho_{\mathrm{MV}}^\lambda$ (Example 2.1).

We now demonstrate that the $m$-moment sketch $\psi_\mu^m$ introduced in Example 5.2 suffices for risk-sensitive learning with respect to a large collection of risk measures.

**Proposition A.3.** *Suppose $\psi = (\psi_1, \ldots, \psi_m)$ is a Bellman-closed sketch and for each $i = 1, \ldots, m$, $\exists f_i : \mathbb{R} \to \mathbb{R}$ such that for each $\nu \in \mathscr{P}_\psi(\mathbb{R})$, $\psi_i(\nu) = \mathbb{E}_{Z \sim \nu}[f_i(Z)]$. Then any risk measure $\rho$ which can be planned for exactly using a proper $\psi$ equivalent model can be planned for exactly using a proper $\psi_\mu^m$ equivalent model.*

## A.2   On the continuity of statistical functionals

We define a sketch $\psi$ to be continuous if whenever a sequence $(\nu_n)_{n \geq 0} \subseteq \mathscr{P}_\psi(\mathbb{R})$ converges to $\nu \in \mathscr{P}_\psi(\mathbb{R})$, we have that $\psi(\nu_n)$ converges to $\psi(\nu)$. We now formalize this notion. We will use various quantities from topology, Munkres (2000) may be used a reference for further details.

We will write $C_b(\mathbb{R})$ for the set of bounded continuous functions from $\mathbb{R}$ to $\mathbb{R}$. We recall that a sequence of measures $(\nu_n)_{n \geq 0} \subseteq \mathscr{P}(\mathbb{R})$ converges weakly to $\nu \in \mathscr{P}(\mathbb{R})$ if

$$\int f d\nu_n \to \int f d\nu,$$

for all $f \in C_b(\mathbb{R})$. We refer to the topology induced by this convergence as the *weak* topology on $\mathscr{P}(\mathbb{R})$ (to be precise, specifying convergence is sufficient to induce the entire topology since this topology is metrizable).

With this definition, we endow $\mathscr{P}(\mathbb{R})^{\mathcal{X}}$ with the product topology generated by the weak topology on $\mathscr{P}(\mathbb{R})$. Then by definition of the product topology, a sequence $(\eta_n)_{n \geq 0} \subseteq \mathscr{P}(\mathbb{R})^{\mathcal{X}}$ converges to $\eta \in \mathscr{P}(\mathbb{R})^{\mathcal{X}}$ if and only if for each $x \in \mathcal{X}$, $(\eta_n(x))_{n \geq 0}$ converges weakly to $\eta(x)$ (note this is weak convergence in $\mathscr{P}(\mathbb{R})$).

We can now define a sketch $\psi : \mathscr{P}_\psi(\mathbb{R}) \to \mathbb{R}^m$ to be (sequentially) continuous if whenever a sequence $(\eta_n)_{n \geq 0} \subseteq \mathscr{P}_\psi(\mathbb{R})^{\mathcal{X}}$ converges to $\eta \in \mathscr{P}_\psi(\mathbb{R})^{\mathcal{X}}$ with the topology we defined above, we have that $\psi(\eta_n)$ converges to $\psi(\eta)$ in the usual topology on $\mathbb{R}^m$.

To see that this continuity of $\psi$ implies convergence of the iterates $(s_k)_{k \geq 0}$ to $s_\psi^\pi$, we can recall that we had $s_k = \psi(\eta_k)$, where $\eta_0 = \iota(s_0)$ and $\eta_{k+1} = \mathcal{T}^\pi \eta_k$. The sequence $(\eta_k)_{k \geq 0}$ converges to $\eta^\pi$ in the weak product topology on $\mathscr{P}(\mathbb{R})^{\mathcal{X}}$ (Bellemare et al., 2023), which then immediately gives that if $\psi$ is continuous as above, $\psi(\eta_k) \to \psi(\eta^\pi)$, and hence $s_k \to s_\psi^\pi$.

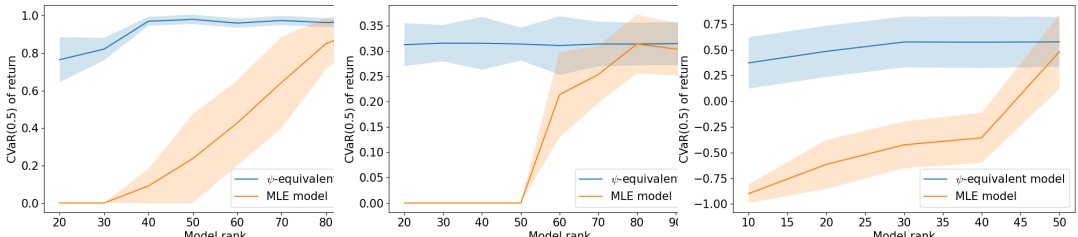

Figure 4: CVaR of returns for policies learnt using a $\psi$-equivalent model and a MLE-based model as the number of distracting dimensions increases (**left**: Four Rooms, **middle**: Frozen Lake, **right**: Windy Cliffs.

## A.3   Additional empirical results

Similar to Figure 3, we performed an additional experiment in the tabular domains, where we limit the model capacity by constraining the rank of transition matrix being estimated. In this regime, the model should ideally use its limited capacity to model the aspects of the environment which are most important for the decision problem. We report the results of this experiment in Figure 4.

## B   Learning statistical functional equivalent models

To learn a model $\tilde{m} \in \mathcal{M}^k(\Pi, \mathcal{I})$, we can define the loss of a model as the total deviation from the definition of $\mathcal{M}^k(\Pi, \mathcal{I})$. To this end, we define

$$\mathcal{L}^{k,p}_{\psi,\Pi,\mathcal{I}}(\tilde{m}) \triangleq \sum_{\pi \in \Pi} \sum_{s \in \mathcal{I}} \left\| (\psi \mathcal{T}^\pi \iota)^k \, s - (\psi \mathcal{T}^\pi_{\tilde{m}} \iota)^k \, s \right\|_p^p ,$$

where for $s = (s_1, \ldots, s_m) \in \mathbb{R}^m$, $\|s\|_p^p = \sum_{i=1}^m |s_i|^p$. If the Bellman operator for $\psi$ exists and is readily available, we can alternatively define the loss working directly on statistics, without needing to impute into distribution space:

$$\mathcal{L}^{k,p}_{\psi,\Pi,\mathcal{I}}(\tilde{m}) = \sum_{\pi \in \Pi} \sum_{s \in \mathcal{I}} \left\| (\mathcal{T}^\pi_\psi)^k s - (\mathcal{T}^\pi_{\psi,\tilde{m}})^k s \right\|_p^p .$$

To learn a proper value equivalent model, Grimm et al. (2021) leverages the fact that for any $k \in \mathbb{N}$ the proper value equivalent class can be deconstructed into an intersection of one proper value equivalent class per policy it matches over:

$$\mathcal{M}^\infty(\Pi) = \bigcap_{\pi \in \Pi} \mathcal{M}^k (\{\pi\}, \{V^\pi\}),$$

so that minimizing $\left| V^\pi - (T^\pi_{\tilde{m}})^k V^\pi \right|$ across all $\pi \in \Pi$ is sufficient to learn a model in $\mathcal{M}^\infty(\Pi)$. We now show that the same argument can be used to learn proper statistical functional equivalent models.

**Proposition B.1.** *If $\psi$ is both continuous and Bellman-closed and $\iota$ is exact, for any $k \in \mathbb{N}$ and $\Pi \subseteq \mathbb{\Pi}$, it holds that*

$$\mathcal{M}^\infty_\psi(\Pi) = \bigcap_{\pi \in \Pi} \mathcal{M}^k_\psi \left( \{\pi\}, \{s^\pi_\psi\} \right) .$$

With this in mind, we can now propose a loss for learning proper statistical functional equivalent models.

**Definition B.2.** Let $\psi$ be a sketch and $\iota$ an imputation strategy. We define the loss for learning a proper $\psi$ equivalent model as

$$\mathcal{L}^{k,p}_{\psi,\Pi,\infty}(\tilde{m}) \triangleq \sum_{\pi \in \Pi} \left\| s^\pi_\psi - (\psi \mathcal{T}^\pi_{\tilde{m}} \iota)^k \, s^\pi_\psi \right\|_p^p .$$

If $\psi$ is Bellman-closed this loss can be written in terms of its Bellman operator, given by

$$\mathcal{L}^{k,p}_{\psi,\Pi,\infty}(\tilde{m}) = \sum_{\pi \in \Pi} \left\| s^\pi_\psi - (\mathcal{T}^\pi_{\psi,\tilde{m}})^k s^\pi_\psi \right\|_p^p .$$

## C Proofs

### C.1 Section 3 Proofs

**Proposition 3.1.** *Let $\rho$ be a risk measure such that for any MDP and any set $\mathcal{M}$ of models, a policy optimal for $\rho$ for any $\tilde{m} \in \mathcal{M}^\infty(\Pi)$ is optimal in the true MDP. Then $\rho$ must be risk-neutral, in the sense that there exists an increasing function $g : \mathbb{R} \to \mathbb{R}$ such that $\rho(\nu) = g(\mathbb{E}_{Z \sim \nu}[Z])$.*

*Proof.* To begin, note that this condition implies that for probability measures $\nu_1, \nu_2$ with $\mathbb{E}_{Z_1 \sim \nu_1}[Z_1] = \mathbb{E}_{Z_2 \sim \nu_2}[Z_2]$, it must hold that $\rho(\nu_1) = \rho(\nu_2)$. To see why, suppose this weren't the case. Then there exists a pair of probability measures $\nu_1, \nu_2 \in \mathscr{P}(\mathbb{R})$ such that $\mathbb{E}_{Z_1 \sim \nu_1}[Z_1] = \mathbb{E}_{Z_2 \sim \nu_2}[Z_2]$, and $\rho(\nu_1) < \rho(\nu_2)$. Then let us construct an MDP $M^*$ where $\mathcal{X} = \{x\}$, $\mathcal{A} = \{a, b\}$, $\gamma = 0$, $\mathcal{R}(x, a) = \nu_1$, $\mathcal{R}(x, b) = \nu_2$ ($\mathcal{P}$ is defined implicitly since there is a single state). Moreover let us define a second MDP $\tilde{M}$ defined by $\mathcal{X} = \{x\}$, $\mathcal{A} = \{a, b\}$, $\gamma = 0$, $\mathcal{R}(x, a) = \mathbb{E}_{Z_1 \sim \nu_1}[Z_1]$, $\mathcal{R}(x, a) = \mathbb{E}_{Z_2 \sim \nu_2}[Z_2]$. Then it is immediate to see that $M^*$ and $\tilde{M}$ are proper value equivalent, however the policy $\pi^a$ defined by $\pi^a(a \mid x) = 1$ is optimal in $\tilde{M}$, but not in $M^*$, contradicting the original statement.

This in turn implies that $\rho(\nu) = f(\mathbb{E}_{Z \sim \nu}[Z])$ for some function $f$. It remains to show that $f$ must be increasing. To see this, suppose not: then there exists $\mu_1, \mu_2 \in \mathscr{P}(\mathbb{R})$ such that $\mathbb{E}_{Z_1 \sim \nu_1}[Z_1] > \mathbb{E}_{Z_2 \sim \nu_2}[Z_2]$ but $\rho(\nu_1) < \rho(\nu_2)$. Then we can construct another pair of MDPs: $M^*$ is defined by setting $\mathcal{X} = \{x\}$, $\mathcal{A} = \{a, b\}$, $\gamma = 0$, $\mathcal{R}(x, a) = \nu_1$, $\mathcal{R}(x, b) = \nu_2$, and $\tilde{M}$ is defined by $\mathcal{X} = \{x\}$, $\mathcal{A} = \{a, b\}$, $\gamma = 0$, $\mathcal{R}(x, a) = \mathbb{E}_{Z_1 \sim \nu_1}[Z_1]$, $\mathcal{R}(x, a) = \mathbb{E}_{Z_2 \sim \nu_2}[Z_2]$. Then once again we can see that $M^*$ and $\tilde{M}$ are proper value equivalent, but the the policy $\pi^a$ defined by $\pi^a(a \mid x) = 1$ is optimal in $\tilde{M}$, but not in $M^*$, giving us our contradiction.

Hence we must have that $\rho(\nu) = f(\mathbb{E}_{Z \sim \nu}[Z])$ for some increasing function $f$, as desired.

$\square$

**Proposition 3.2.** *Let $\rho$ be an $(\varepsilon, \delta)$-strictly risk-sensitive spectral risk measure, and suppose that rewards are almost surely bounded by $R_{max}$. Then there exists an MDP with a proper value equivalent model $\tilde{m}$ with the following property: letting $\pi_\rho^*$ be an optimal policy for $\rho$ in the original MDP, and $\tilde{\pi}_\rho^*$ an optimal policy for $\rho$ in $\tilde{m}$, we have*

$$\inf_{x \in \mathcal{X}} \left\{ \rho\big(\eta^{\pi_\rho^*}(x)\big) - \rho\big(\eta^{\tilde{\pi}_\rho^*}(x)\big) \right\} \geq \frac{R_{max}}{1 - \gamma} \varepsilon(1 - \delta(1 - \varepsilon)).$$

*Proof.* Let $\varphi$ be the function which $\rho$ corresponds to (so that $\rho(\mu) = \int_0^1 F_\mu^{-1}(u) \, \varphi(u) \, du$). As $\rho$ is strictly risk-sensitive, let $\varepsilon, \delta$ be such that $\varphi(\varepsilon) \leq \delta$. Next, note that since $\varphi$ is constrained to be positive, non-increasing, and integrating to 1, we have that

$$\int_0^1 F_\mu^{-1}(u) \, \varphi(u) \, du = \int_0^{1-\varepsilon} F_\mu^{-1}(u) \, \varphi(u) \, du + \int_{1-\varepsilon}^1 F_\mu^{-1}(u) \, \varphi(u) \, du$$

$$\leq \frac{1}{1 - \varepsilon} \int_0^1 F_\mu^{-1}(u) \, \mathbb{1}_{[0, 1-\varepsilon]} \, du + \delta \int_0^1 F_\mu^{-1}(u) \, du$$

$$= \frac{1}{1 - \varepsilon} \int_0^{1-\varepsilon} F_\mu^{-1}(u) \, du + \delta \int_{1-\varepsilon}^1 F_\mu^{-1}(u) \, du.$$

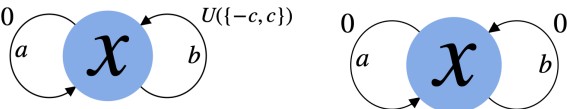

Figure 5: An MDP $m^*$ (left) and a proper value equivalent model $\tilde{m}$ (right).

Let us now consider the MDPs $m^*$ and $\tilde{m}$ as given in Figure 5. Following Example 2.10 in Bellemare et al. (2023), we have that $\eta^{\pi^b}(x) = U([-2c, 2c])$, so that $F_{\eta^{\pi^b}(x)}^{-1}(u) = 4cu - 2c$. We can use this

to calculate

$$\rho(\eta^{\pi^b}(x)) = \int_0^1 F_\mu^{-1}(u)\,\varphi(u)\,du$$

$$\leq \frac{1}{1-\varepsilon}\int_0^{1-\varepsilon} F_{\eta^{\pi^b}(x)}^{-1}(u)\,du + \delta\int_{1-\varepsilon}^1 F_{\eta^{\pi^b}(x)}^{-1}(u)\,du$$

$$= \frac{1}{1-\varepsilon}\int_0^{1-\varepsilon} (4cu - 2c)\,du + \delta\int_{1-\varepsilon}^1 (4cu - 2c)\,du$$

$$= -2c\varepsilon(1 - \delta(1-\varepsilon)).$$

With this calculation done, we can remark that $\pi^a$ is optimal in $m^*$, as we have $\rho(\pi^a(x)) = 0$. Moreover, $\pi^b$ is an optimal policy in $\tilde{m}$, as $\rho(\pi^a_{\tilde{m}}(x)) = \rho(\pi^b_{\tilde{m}}(x)) = 0$.

We can then see that
$$\rho(\pi^a(x)) - \rho(\pi^b(x)) \geq 2c\varepsilon(1 - \delta(1 - \varepsilon)),$$
which completes the proof. $\qquad\square$

## C.2   Section 4 Proofs

**Theorem 4.3.** *Let $\rho$ be any risk measure. Then an optimal policy with respect to $\rho$ in $\tilde{m} \in \mathcal{M}_{\mathrm{dist}}^\infty(\Pi)$ is optimal with respect to $\rho$ in $m^*$.*

*Proof.* Let $\pi_\rho^*$ be an optimal policy for $\rho$ in $m^*$, and let $\tilde{\pi}_\rho^*$ be an optimal policy for $\rho$ in $\tilde{m}$. For contradiction, suppose that $\tilde{\pi}_\rho^*$ is not optimal in $m^*$. Then for all $x \in \mathcal{X}$ we have that

$$\rho(\eta^{\tilde{\pi}_\rho^*}(x)) \leq \rho(\eta^{\pi_\rho^*}(x)),$$

and for at least one $x \in \mathcal{X}$ we have

$$\rho(\eta^{\tilde{\pi}_\rho^*}(x)) < \rho(\eta^{\pi_\rho^*}(x)).$$

Let us choose this $x$, and note that this implies

$$\rho\left(\eta^{\tilde{\pi}_\rho^*}(x)\right) < \rho\left(\eta^{\pi_\rho^*}(x)\right)$$

$$\iff \quad \rho\left(\eta^{\tilde{\pi}_\rho^*}_{\tilde{m}}(x)\right) < \rho\left(\eta^{\pi_\rho^*}_{\tilde{m}}(x)\right),$$

since by assumption of $\tilde{m} \in \mathcal{M}_{\mathrm{dist}}^\infty(\Pi)$ we have that $\eta^\pi = \eta^\pi_{\tilde{m}}$ for any $\pi \in \Pi$. But this contradicts the assumption that $\tilde{\pi}_\rho^*$ was optimal for $\rho$ in $\tilde{m}$, and we are complete. $\qquad\square$

## C.3   Section 5 Proofs

**Proposition 5.5.** *If $\psi$ is Bellman-closed and $\iota$ is exact, we have that*
$$\mathcal{M}_\psi(\Pi, \mathcal{I}) = \left\{\tilde{m} \in \mathcal{M} : \mathcal{T}_\psi^\pi s = \mathcal{T}_{\psi,\tilde{m}}^\pi s,\ \forall \pi \in \Pi, \forall s \in \mathcal{I}\right\}.$$

*Proof.* Recall that

$$\mathcal{M}_\psi(\Pi, \mathcal{I}) = \left\{\tilde{m} \in \mathcal{M} : \psi\left(\mathcal{T}^\pi \iota(s)\right) = \psi\left(\mathcal{T}_{\tilde{m}}^\pi \iota(s)\right)\ \forall \pi \in \Pi, s \in \mathcal{I}\right\}.$$

Note that $\psi\left(\mathcal{T}^\pi \iota(s)\right) = \mathcal{T}_\psi^\pi \psi(\iota(s))$ since $\psi$ is Bellman-closed, and since $\iota$ is exact we have that $\psi(\iota(s)) = s$. Combining these we have that $\psi\left(\mathcal{T}^\pi \iota(s)\right) = \mathcal{T}_\psi^\pi s$, which then gives us equality of the sets as desired. $\qquad\square$

**Definition C.1.** *Let $(A_k)_{k=1}^\infty$ be a sequence of sets. Then we have*

$$\liminf_{k\to\infty} A_k = \bigcup_{k\geq 1}\bigcap_{j\geq k} A_j, \text{ and } \limsup_{k\to\infty} A_k = \bigcap_{k\geq 1}\bigcup_{j\geq k} A_j.$$

If both of these sets are equal, then we say that $\lim_{k\to\infty} A_k$ exists and is equal to that common set.

**Proposition 5.7.** *If $\psi$ is both continuous and Bellman-closed and $\iota$ is exact, then*

$$\lim_{k\to\infty}\mathcal{M}_\psi^k(\Pi, \mathcal{I}) = \mathcal{M}_\psi^\infty(\Pi), \text{ for any } \mathcal{I} \subseteq I_\psi^\mathcal{X}.$$

*Proof.* We can begin by recalling that for $k > 0$,

$$\mathcal{M}_\psi^k(\Pi, \mathcal{I}) = \left\{ \tilde{m} \in \mathcal{M} : (\psi \mathcal{T}^\pi \iota)^k s = (\psi \mathcal{T}_{\tilde{m}}^\pi \iota)^k s \ \ \forall \pi \in \Pi, s \in \mathcal{I} \right\},$$

We can also note that if $\tilde{m} \in \mathcal{M}_\psi^k(\Pi, \mathcal{I})$, then $\tilde{m} \in \mathcal{M}_\psi^{nk}(\Pi, \mathcal{I})$ for $n > 0$, since if $(\psi \mathcal{T}^\pi \iota)^k s = (\psi \mathcal{T}_{\tilde{m}}^\pi \iota)^k s$, then by setting both sides to the power $n$ we have that $(\psi \mathcal{T}^\pi \iota)^{nk} s = (\psi \mathcal{T}_{\tilde{m}}^\pi \iota)^{nk} s$. This implies that $\mathcal{M}_\psi^k(\Pi, \mathcal{I}) \subseteq \mathcal{M}_\psi^{nk}(\Pi, \mathcal{I})$. Since this is true for any $n$, we can set $n \to \infty$ to obtain

$$\mathcal{M}_\psi^k(\Pi, \mathcal{I}) \subseteq \left\{ \tilde{m} \in \mathcal{M} : \lim_{n \to \infty} (\psi \mathcal{T}^\pi \iota)^{nk} s = \lim_{n \to \infty} (\psi \mathcal{T}_{\tilde{m}}^\pi \iota)^{nk} s \ \ \forall \pi \in \Pi, s \in \mathcal{I} \right\}.$$

Since $\iota$ is exact and $\psi$ is Bellman-closed, we have that for any $n \geq 0$, $(\psi \mathcal{T}^\pi \iota)^{nk} s = \psi((\mathcal{T}^\pi)^{nk} \iota(s))$ (Proposition 8.9 in Bellemare et al. (2023)). Since $\psi$ is continuous, we have that $\psi((\mathcal{T}^\pi)^{nk} \iota(s)) \to s^\pi$ as $n \to \infty$ (justification for this can be found in Appendix A.2). We can then use this to rewrite the above as

$$\mathcal{M}_\psi^k(\Pi, \mathcal{I}) \subseteq \left\{ \tilde{m} \in \mathcal{M} : s_\psi^\pi = s_{\psi, \tilde{m}}^\pi \ \ \forall \pi \in \Pi, s \in \mathcal{I} \right\} = \mathcal{M}^\infty(\Pi).$$

This immediately gives us that

$$\bigcup_{j \geq k} \mathcal{M}_\psi^j(\Pi, \mathcal{I}) \subseteq \mathcal{M}^\infty(\Pi).$$

Since this expression is independent of $k$, we can take the intersection over all $k$ to see that

$$\limsup_{k \to \infty} \mathcal{M}_{\mathrm{dist}}^k(\Pi) = \bigcap_{k \geq 1} \bigcup_{j \geq k} \mathcal{M}_\psi^j(\Pi, \mathcal{I}) \subseteq \mathcal{M}^\infty(\Pi).$$

Moreover it is immediate to see that

$$\mathcal{M}^\infty(\Pi) \subseteq \bigcap_{k \geq 1} \bigcup_{j \geq k} \mathcal{M}_\psi^j(\Pi, \mathcal{I}),$$

which together gives us

$$\limsup_{k \to \infty} \mathcal{M}_{\mathrm{dist}}^k(\Pi) = \mathcal{M}_{\mathrm{dist}}^\infty(\Pi).$$

We now focus on the limit inferior. We take $k > 0$, and see that

$$\bigcap_{j \geq k} \mathcal{M}_\psi^j(\Pi, \mathcal{I}) = \left\{ \tilde{m} \in \mathcal{M} : (\psi \mathcal{T}^\pi \iota)^j s = (\psi \mathcal{T}_{\tilde{m}}^\pi \iota)^j s \ \ \forall j \geq k, \pi \in \Pi, s \in \mathcal{I} \right\}$$

$$\subseteq \left\{ \tilde{m} \in \mathcal{M} : \lim_{j \to \infty} (\psi \mathcal{T}^\pi \iota)^j s = \lim_{j \to \infty} (\psi \mathcal{T}_{\tilde{m}}^\pi \iota)^j s \ \ \pi \in \Pi, s \in \mathcal{I} \right\}.$$

As argued above for the limit superior, we have that $\lim_{j \to \infty} \psi((\mathcal{T}^\pi)^j \iota(s)) = s_\psi^\pi$. Using this fact in the original expression above, we have

$$\left\{ \tilde{m} \in \mathcal{M} : \lim_{j \to \infty} (\psi \mathcal{T}^\pi \iota)^j s = \lim_{j \to \infty} (\psi \mathcal{T}_{\tilde{m}}^\pi \iota)^j s \ \ \pi \in \Pi, s \in \mathcal{I} \right\} = \left\{ \tilde{m} \in \mathcal{M} : s_{\psi, \tilde{m}}^\pi = s_\psi^\pi \ \ \pi \in \Pi, s \in \mathcal{I} \right\}.$$

Conversely, it is immediate to see that

$$\left\{ \tilde{m} \in \mathcal{M} : s_{\psi, \tilde{m}}^\pi = s_\psi^\pi \ \ \pi \in \Pi, s \in \mathcal{I} \right\} \subseteq \left\{ \tilde{m} \in \mathcal{M} : (\psi \mathcal{T}^\pi \iota)^j s = (\psi \mathcal{T}_{\tilde{m}}^\pi \iota)^j s \ \ \forall j \geq k, \pi \in \Pi, s \in \mathcal{I} \right\},$$

so that we can combine with our work above and conclude that

$$\bigcap_{j \geq k} \mathcal{M}_\psi^j(\Pi, \mathcal{I}) \subseteq \left\{ \tilde{m} \in \mathcal{M} : s_{\psi, \tilde{m}}^\pi = s_\psi^\pi \ \ \pi \in \Pi, s \in \mathcal{I} \right\} = \mathcal{M}_{\mathrm{dist}}^\infty(\Pi).$$

Since this expression is independent of $k$, we can take the union over $k$ to obtain

$$\liminf_{k \to \infty} \mathcal{M}_\psi^k(\Pi, \mathcal{I}) = \bigcup_{k \geq 1} \bigcap_{j \geq k} \mathcal{M}_\psi^j(\Pi, \mathcal{I}) \subseteq \mathcal{M}_{\mathrm{dist}}^\infty(\Pi).$$

Moreover it is immediate to see that

$$\mathcal{M}_{\mathrm{dist}}^\infty(\Pi) \subseteq \bigcup_{k \geq 1} \bigcap_{j \geq k} \mathcal{M}_\psi^j(\Pi, \mathcal{I}),$$

which together give

$$\liminf_{k\to\infty} \mathcal{M}_\psi^k(\Pi, \mathcal{I}) = \mathcal{M}_{\text{dist}}^\infty(\Pi).$$

Since the limit inferior and limit superior are equal, we have the existence of the limit

$$\lim_{k\to\infty} \mathcal{M}_\psi^k(\Pi, \mathcal{I}) = \mathcal{M}_{\text{dist}}^\infty(\Pi).$$

$\square$

**Proposition 5.9.** *Suppose $\iota$ is injective, $\Pi \subseteq \bar{\Pi}$, $\mathcal{I} \subseteq I_\psi^{\mathcal{X}}$, and let $\mathcal{D}_{\mathcal{I}} = \{\iota(s) : s \in \mathcal{I}\} \subseteq \mathscr{P}_\psi(\mathbb{R})^{\mathcal{X}}$. Then*

$$\mathcal{M}_\psi(\Pi, \mathcal{I}) = \left\{\tilde{m} \in \mathcal{M} : \Pi_{\mathscr{F}_\psi} \mathcal{T}^\pi \eta = \Pi_{\mathscr{F}_\psi} \mathcal{T}_{\tilde{m}}^\pi \eta, \forall \pi \in \Pi, \forall \eta \in \mathcal{D}_{\mathcal{I}}\right\},$$

*and*

$$\mathcal{M}_\psi^\infty(\Pi) = \left\{\tilde{m} \in \mathcal{M} : \Pi_{\mathscr{F}_\psi} \eta^\pi = \Pi_{\mathscr{F}_\psi} \eta_{\tilde{m}}^\pi, \ \forall \pi \in \Pi\right\}.$$

*Proof.* We can write out

$$\mathcal{M}_\psi(\Pi, \mathcal{I}) = \left\{\tilde{m} \in \mathcal{M} : \psi\left(\mathcal{T}^\pi \iota(s)\right) = \psi\left(\mathcal{T}_{\tilde{m}}^\pi \iota(s)\right) \ \forall \pi \in \Pi, s \in \mathcal{I}\right\}$$

$$= \left\{\tilde{m} \in \mathcal{M} : \psi\left(\mathcal{T}^\pi \eta\right) = \psi\left(\mathcal{T}_{\tilde{m}}^\pi \eta\right) \ \forall \pi \in \Pi, s \in \mathcal{D}\right\}$$

$$= \left\{\tilde{m} \in \mathcal{M} : \iota(\psi\left(\mathcal{T}^\pi \eta\right)) = \iota(\psi\left(\mathcal{T}_{\tilde{m}}^\pi \eta\right)) \ \forall \pi \in \Pi, s \in \mathcal{D}\right\}$$

$$= \left\{\tilde{m} \in \mathcal{M} : \Pi_{\mathscr{F}_\psi} \mathcal{T}^\pi \eta = \Pi_{\mathscr{F}_\psi} \mathcal{T}_{\tilde{m}}^\pi \eta \ \forall \pi \in \Pi, s \in \mathcal{D}\right\},$$

where the second to last inequality follows from the injectivity of $\iota$. Similarly we have that

$$\mathcal{M}_\psi^\infty(\Pi, \mathcal{I}) = \left\{\tilde{m} \in \mathcal{M} : \psi\left(\eta^\pi\right) = \psi\left(\eta_{\tilde{m}}^\pi\right) \ \forall \pi \in \Pi\right\}$$

$$= \left\{\tilde{m} \in \mathcal{M} : \iota(\psi\left(\eta^\pi\right)) = \psi\left(\iota(\eta_{\tilde{m}}^\pi)\right) \forall \pi \in \Pi\right\}$$

$$= \left\{\tilde{m} \in \mathcal{M} : \Pi_{\mathscr{F}_\psi} \eta^\pi = \Pi_{\mathscr{F}_\psi} \eta_{\tilde{m}}^\pi \ \forall \pi \in \Pi\right\},$$

where the second equality follows by injectivity of $\iota$. $\square$

**Proposition 5.10.** *Let $\rho$ be a risk measure and let $\psi = (\psi_1, \dots, \psi_m)$ be a sketch, and suppose that $\rho$ is in the span of $\psi$, in the sense that there exists $\alpha_0, \dots \alpha_m \in \mathbb{R}$ such that for all $\nu \in \mathscr{P}_\psi(\mathbb{R}) \cap \mathscr{P}_\rho(\mathbb{R})$, $\rho(\nu) = \sum_{i=1}^m \alpha_i \psi_i(\nu) + \alpha_0$. Then any optimal policy with respect to $\rho$ in $\tilde{m} \in \mathcal{M}_\psi^\infty(\bar{\Pi})$ is optimal with respect to $\rho$ in $m^*$.*

*Proof.* Let $\pi_\rho^*$ be an optimal policy for $\rho$ in $m^*$, and let $\tilde{\pi}_\rho^*$ be an optimal policy for $\rho$ in $\tilde{m}$. For contradiction, suppose that $\tilde{\pi}_\rho^*$ is not optimal in $m^*$. Then for all $x \in \mathcal{X}$ we have that

$$\rho(\eta^{\tilde{\pi}_\rho^*}(x)) \leq \rho(\eta^{\pi_\rho^*}(x)),$$

and for at least one $x \in \mathcal{X}$ we have

$$\rho(\eta^{\tilde{\pi}_\rho^*}(x)) < \rho(\eta^{\pi_\rho^*}(x)).$$

Let us choose this $x$, and note that this implies

$$\rho\left(\eta^{\tilde{\pi}_\rho^*}(x)\right) < \rho\left(\eta^{\pi_\rho^*}(x)\right)$$

$$\iff \quad \sum_{i=1}^m \alpha_i \psi_i\left(\eta^{\tilde{\pi}_\rho^*}(x)\right) < \sum_{i=1}^m \alpha_i \psi_i\left(\eta^{\pi_\rho^*}(x)\right)$$

$$\iff \quad \sum_{i=1}^m \alpha_i \psi_i\left(\eta_{\tilde{m}}^{\tilde{\pi}_\rho^*}(x)\right) < \sum_{i=1}^m \alpha_i \psi_i\left(\eta_{\tilde{m}}^{\pi_\rho^*}(x)\right)$$

$$\iff \quad \rho\left(\eta_{\tilde{m}}^{\tilde{\pi}_\rho^*}(x)\right) < \rho\left(\eta_{\tilde{m}}^{\pi_\rho^*}(x)\right),$$

which contradicts the assumption that $\tilde{\pi}_\rho^*$ was optimal for $\rho$ in $\tilde{m}$. $\qquad\square$

### C.4 Appendix Proofs

**Proposition A.3.** *Suppose $\psi = (\psi_1, \ldots, \psi_m)$ is a Bellman-closed sketch and for each $i = 1, \ldots, m$, $\exists f_i : \mathbb{R} \to \mathbb{R}$ such that for each $\nu \in \mathscr{P}_\psi(\mathbb{R})$, $\psi_i(\nu) = \mathbb{E}_{Z \sim \nu}[f_i(Z)]$. Then any risk measure $\rho$ which can be planned for exactly using a proper $\psi$ equivalent model can be planned for exactly using a proper $\psi_\mu^m$ equivalent model.*

*Proof.* This proof relies on a theorem introduced by Rowland et al. (2019), which we restate here.

**Theorem C.2** (Rowland et al. (2019))**.** *Let $\psi = (\psi_1, \ldots, \psi_m)$ be a Bellman closed sketch such that for each $i = 1, \ldots, m$, there exist $f_i : \mathbb{R} \to \mathbb{R}$ such that for all $\nu \in \mathscr{P}_\psi(\mathbb{R})$, $\psi_i(\nu) = \mathbb{E}_{Z \sim \nu}[f_i(Z)]$. Then there exists real numbers $(b_{ij})_{i,j=1}^m$ such that for all $\nu \in \mathscr{P}_\psi(\mathbb{R})$,*

$$\psi_i(\nu) = \sum_{j=1}^m b_{ij} \mu_j(\nu) + b_{i0},$$

*where $\mu_j$ is the jth moment functional (Example 5.2).*

Next, let us suppose that $\rho$ can be planned for optimally by any $\tilde{m} \in \mathcal{M}_\psi^\infty(\Pi)$, then there must exist $(\alpha_i)_{i=1}^m$ such that for all $\nu \in \mathscr{P}_\psi(\mathbb{R}) \bigcap \mathscr{P}_\rho(\mathbb{R})$, $\rho(\nu) = \sum_{i=1}^m \alpha_i \psi_i(\nu)$. Using the coefficients $(b_{ij})_{i,j=1}^m$ introduced in the theorem statement above, we have that

$$\rho(\nu) = \sum_{i=1}^m \alpha_i(\psi_i(\nu)$$

$$= \sum_{i=1}^m \alpha_i \left( \sum_{j=1}^m b_{ij} \mu_j(\nu) + b_{i0} \right)$$

$$= \sum_{i=1}^m \alpha_i \sum_{j=1}^m b_{ij} \mu_j(\nu) + \sum_{i=1}^m \alpha_i b_{i0}$$

$$= \sum_{j=1}^m \beta_j \mu_j(\nu) + \beta_0,$$

where $\beta_j = \sum_{i=1}^m \alpha_i b_{ij}$ for $j = 0, \ldots, m$. We can then apply Proposition 5.10, and we are complete.

$\square$

**Proposition B.1.** *If $\psi$ is both continuous and Bellman-closed and $\iota$ is exact, for any $k \in \mathbb{N}$ and $\Pi \subseteq \mathbb{\Pi}$, it holds that*

$$\mathcal{M}_\psi^\infty(\Pi) = \bigcap_{\pi \in \Pi} \mathcal{M}_\psi^k\left(\{\pi\}, \{s_\psi^\pi\}\right).$$

*Proof.* We begin by rewriting the definition

$$\mathcal{M}_\psi^\infty(\Pi) = \left\{ \tilde{m} \in \mathcal{M} : s_{\psi,\tilde{m}}^\pi = s_\psi^\pi \text{ for all } \pi \in \Pi \right\}$$

$$= \bigcap_{\pi \in \Pi} \left\{ \tilde{m} \in \mathcal{M} : s_{\psi,\tilde{m}}^\pi = s_\psi^\pi \right\}$$

$$= \bigcap_{\pi \in \Pi} \mathcal{M}_\psi^\infty(\{\pi\}).$$

Next, note that for any $\pi$ and any $k \in \mathbb{N}$ we have $\mathcal{M}_\psi^\infty(\{\pi\}) \subseteq \mathcal{M}_\psi^k(\{\pi\}, \{s_\psi^\pi\})$, since if $s_\psi^\pi = s_{\psi,\tilde{m}}^\pi$ we can write out

$$s_\psi^\pi = s_{\psi,\tilde{m}}^\pi$$
$$\implies \quad (\mathcal{T}_{\psi,\tilde{m}}^\pi)^k s_\psi^\pi = (\mathcal{T}_{\psi,\tilde{m}}^\pi)^k s_{\psi,\tilde{m}}^\pi$$
$$\implies \quad (\mathcal{T}_{\psi,\tilde{m}}^\pi)^k s_\psi^\pi = s_{\psi,\tilde{m}}^\pi$$
$$\implies \quad (\mathcal{T}_{\psi,\tilde{m}}^\pi)^k s_\psi^\pi = s_\psi^\pi$$
$$\implies \quad (\mathcal{T}_{\psi,\tilde{m}}^\pi)^k s_\psi^\pi = (\mathcal{T}_\psi^\pi)^k s_\psi^\pi,$$

and hence $\tilde{m} \in \mathcal{M}_\psi^k(\{\pi\}, \{s_\psi^\pi\})$. Conversely, if we let $\tilde{m} \in \mathcal{M}_\psi^k(\{\pi\}, \{s_\psi^\pi\})$, we can write out

$$(\mathcal{T}_{\psi,\tilde{m}}^\pi)^k s_\psi^\pi = (\mathcal{T}_\psi^\pi)^k s_\psi^\pi$$
$$\implies \quad (\mathcal{T}_{\psi,\tilde{m}}^\pi)^k s_\psi^\pi = s_\psi^\pi$$
$$\implies \quad (\mathcal{T}_{\psi,\tilde{m}}^\pi)^{2k} s_\psi^\pi = (\mathcal{T}_\psi^\pi)^k s_\psi^\pi$$
$$\implies \quad (\mathcal{T}_{\psi,\tilde{m}}^\pi)^{2k} s_\psi^\pi = s_\psi^\pi,$$

which we can repeat $n$ times to obtain $(\mathcal{T}_{\psi,\tilde{m}}^\pi)^{nk} s_\psi^\pi = s_\psi^\pi$. Sending $n \to \infty$ and using the fact that $\psi$ is continuous gives us $s_\psi^\pi = \lim_{n\to\infty} (\mathcal{T}_{\psi,\tilde{m}}^\pi)^{nk} s_\psi^\pi = s_{\psi,\tilde{m}}^\pi$, which then gives us $\tilde{m} \in \mathcal{M}_\psi^k(\{\pi\}, \{s_\psi^\pi\})$. $\qquad\square$

## D   General classes of policies

In Section 2, we considered stationary Markov policies. One can further consider history-dependent policies, which don't have to be stationary nor Markov, but simply measurable with respect to the filtration $\mathcal{F}_t$ given by $\mathcal{F}_t = \sigma\left( \left( \prod_{i=0}^{t-1} (\mathcal{X} \times \mathcal{A}) \right) \times \mathcal{X} \right)$ (where for a collection of sets $A$, $\sigma(A)$ is the $\sigma$-algebra generated by $A$). This is a much larger class of policies, and learning a policy in this class is infeasible in general (Puterman, 2014).

In the risk-neutral setting, the difficulties associated with learning a history-dependent policies can be avoided: for every history-dependent policy, there exists a Markov stationary policy which achieves the same expected return. In particular, no history-dependent policy achieves a return higher than a Markov stationary policy, and thus it suffices to solely consider learning a Markov stationary policy.

Unfortunately, such a result does not exist for the risk-sensitive setting: for a general risk measure, there exists history-dependent policies which achieve a higher objective of return than all Markov stationary policies. Moreover, a Markov stationary policy which is optimal as defined in Section 2 may not exist in general. Despite this negative result, the standard approach in practice is nonetheless to learn an approximately optimal Markov stationary policy (Bäuerle & Ott, 2011; Chow & Ghavamzadeh, 2014; Lim & Malik, 2022), and this is the approach taken in this work as well.

Due to the fact that an optimal policy may not exist, Theorem 4.3 may seem to not generally apply, as it only addresses the case when a Markov stationary optimal policy exists. We now present a weaker version of this theorem, which addresses the case in which such an optimal policy does not exist.

To state the proposition, we first introduce the notion of policy domination. Suppose $\pi_1, \pi_2 \in \Pi$. We say that $\pi_1$ *dominates* $\pi_2$ with respect to $\rho$ if

$$\rho\left(\eta^{\pi_1}(x)\right) \geq \rho\left(\eta^{\pi_2}(x)\right), \forall x \in \mathcal{X}.$$

It is straightforward to see that policy domination provides a partial order over the set of Markov stationary policies $\Pi$. With this in mind, we present the proposition.

**Proposition D.1.** *Let $\rho$ be any risk measure, and let $\pi_1, \pi_2$ be policies such that $\pi_1$ dominates $\pi_2$ with respect to $\rho$ in an approximate model $\tilde{m} \in \mathcal{M}^\infty_{\text{dist}}(\mathbb{\Pi})$. Then $\pi_1$ dominates $\pi_2$ with respect to $\rho$ in $m^*$.*

*Proof.* Let $\pi_1, \pi_2$ satisfy the statement of the proposition. For contradiction, suppose that $\pi_1$ does not dominate $\pi_2$ in $m^*$. Then for all $x \in \mathcal{X}$ we have that

$$\rho(\eta^{\pi_1}(x)) \leq \rho(\eta^{\pi_2}(x)),$$

and for at least one $x \in \mathcal{X}$ we have

$$\rho(\eta^{\pi_1}(x)) < \rho(\eta^{\pi_2}(x)).$$

Let us choose this $x$, and note that this implies

$$\rho(\eta^{\pi_1}(x)) < \rho(\eta^{\pi_2}(x))$$
$$\iff \quad \rho(\eta^{\pi_1}_{\tilde{m}}(x)) < \rho(\eta^{\pi_2}_{\tilde{m}}(x)),$$

since by assumption of $\tilde{m} \in \mathcal{M}^\infty_{\text{dist}}(\mathbb{\Pi})$ we have that $\eta^\pi = \eta^\pi_{\tilde{m}}$ for any $\pi \in \mathbb{\Pi}$. But this contradicts the assumption that $\pi_1$ dominated $\pi_2$ in $\tilde{m}$, and we are complete. $\square$

We note that this proposition should be interpreted as follows: suppose one learns an approximately optimal policy in $\tilde{m} \in \mathcal{M}^\infty_{\text{dist}}(\mathbb{\Pi})$, in the sense that it dominates a set of other candidate policies. Then this policy will be approximately optimal in $m^*$, in the sense that it will still dominate this same set of policies in $m^*$. We note that it is straightforward to adapt Proposition 5.10 in the same way.

# E    Empirical details

We begin with a detailed description of the environments used, followed by details on the compute resources used.

## E.1    Environment descriptions

### E.1.1    Tabular environments

#### Four rooms

We adapt the stochastic four rooms domain used in Grimm et al. (2021) by making certain states risky. In the original domain, an agent attempts to navigate from the start state (bottom left) to the goal state (top right), by moving up, down, left, or right. At each step however, there is a 20% chance that the agent slips and moves in a random direction, rather than the intended one. A reward of 1 is achieved for reaching the goal state, and the reward is 0 elsewhere. We then select certain states to become 'risky' states. These states have the same transition dynamics, but modified reward: if they transition in the intended direction they receive a small, positive reward, and if they transition in a random direction they receive a large negative reward. The rewards are chosen so that the expected reward from a state has a slightly positive expectation, so that risk-neutral policies would pass through the state, but risk-averse ones would not.

#### Windy cliffs

We consider the stochastic adaptation of the cliff walk environment (Sutton & Barto, 2018) as introduced in Bellemare et al. (2023). An agent must walk along a cliff to reach its goal, but at every step, it has a $1/3$ probability of moving in a random direction. A reward of $-1$ is obtained for falling off the cliff, and a reward of 1 is obtained for reaching the goal state.

#### Frozen lake

We use the 8 by 8 frozen lake domain as specified in Brockman et al. (2016). There are four actions corresponding to walking in each direction, however taking an action has a $1/3$ probability of moving in the intended direction, and a $1/3$ probability of moving in each of the perpendicular directions. The agent begins in the top left corner, and attempts to reach the goal at the bottom right corner, at which point the agents receives a reward of 1 and the episode ends. Within the environment there are various holes in the ice, entering a hole will provide a reward of -1 and the episode ends. Episodes will also end after 200 timesteps. Following this, there are 3 possible returns for an episode: $-1$ for

falling in a hole, 1 for reaching the goal, and 0 for reaching the 200 timesteps without reaching a hole state or the goal.

### E.1.2 Option trading environment

We use the option trading environment as implemented in Lim & Malik (2022). In particular, the environment simulates the task of learning a policy of when to exercise American call options. The state space is given as $\mathcal{X} = \mathbb{R}^2$, where for a given $\mathcal{X} \ni x = (p, t)$, $p$ represents the price of the underlying stock, and $t$ represents the time until maturity. The two actions represent holding the stock and executing, and at maturity all options are executed. The training data is generated by assuming that stock prices follows geometric Brownian motion (Li et al., 2009). For evaluation, real stock prices are used, using the data of 10 Dow instruments from 2016-2019.

### E.2 Compute time and infrastructure

For the tabular experiments, each model took roughly 1 hour to train on a single CPU, for an approximate total of 120 CPU hours for the tabular set of experiments. For the option trading experiments, training a policy for a given CVaR level took roughly 40 minutes on a single Tesla P100 GPU on average, for an approximate total of 200 GPU hours for this set of experiments.

## F  Limitations and future work

While we introduced a novel framework and demonstrated strong theoretical and empirical results, our work has limitations, which we now discuss and present as possible directions for future work. The first is investigating how well the statistical functional $\psi$ used can plan for general risk measures not covered by Proposition 5.10, and deriving bounds on its performance. A second is that our theory relies on the set $\mathcal{M}_{\mathrm{dist}}^{\infty}(\mathbb{\Pi})$, while in practice we use $\mathcal{M}_{\mathrm{dist}}^{\infty}(\Pi)$, where $\Pi \subseteq \mathbb{\Pi}$ is a uniformly random subset. Investigating how this affects the theoretical results, along with investigating whether there is a better way to choose the set $\Pi$, are interesting questions in this direction.

