# OpenReview forum: "Distributional Model Equivalence for Risk-Sensitive Reinforcement Learning"
_NeurIPS.cc/2023/Conference — NeurIPS 2023 poster_

### Official Review · Reviewer_B3wZ · 2023-06-21

**Soundness:** 3 good
**Presentation:** 3 good
**Contribution:** 3 good
**Rating:** 7
**Confidence:** 4

**Summary:**

This paper addresses learning models that are sufficient to model the environment, in the sense that optimising for a risk-sensitive objective on that model is equivalent to optimising the risk-sensitive objective on the actual environment. The authors show that value-equivalent models (which match the real environment in expectation) are insufficient in the risk-sensitive case. The authors introduce the distribution equivalence principle, which defines the set of models that induce the same return distribution as the real environment (and therefore the same risk-sensitive values). To relax the assumption that the entire return distribution must be matched, the authors introduce statistical function equivalence, meaning that the models are are equivalent in terms of some statistics of the return (e.g. mean and variance). The authors define loss functions that implement these insights.

**Strengths:**

* I think the paper addresses a novel topic: how best to learn models when the goal is to learn a risk-sensitive policy rather than a standard expected value policy.
* The writing is clear in sections 2-5, and the ideas are well-formalised. I think sections 2-5 are very strong.

**Weaknesses:**

* The paper makes it clear that value-equivalent models fail in the risk-sensitive setting (Proposition 3.2 and experiments), however it fails to motivate why one would use distribution equivalence instead of the standard approach to learning a model - maximum likelihood estimation. Presumably, if the model has enough capacity and is expressive enough, we can expect a model learnt using MLE to learn the correct distributions for distributional equivalence. The paper should explain and demonstrate in which situations the model learnt using the proposed approach results in better performance than MLE model estimate. For example, in the case where model has limited capacity, there is limited data, or the model uses a simplified distribution (such as a Gaussian over successor states), we might expect the proposed approach to work better for risk-sensitive optimisation. However, the paper does not discuss or demonstrate these potential advantages.
* The description of the experiments is unclear - please see my questions.
* In the empirical evaluation, it appears that the authors do not compare against MLE model estimation, which is the most obvious baseline. In Four Rooms/Frozen Lake/Windy Cliffs, the only baseline is the value equivalent model (which is obviously a bad approach for risk-sensitive optimisation). In the option-trading environment, the authors compare against the VE model again, as well as a model-free approach. The authors outperform the model-free approach, but lines 318 and 329 hint that this is because their approach is model-based (and therefore obtains better sample efficiency). Thus, it is unclear if the approach of the authors is better because of the ideas introduced in the paper, or simply because it is model-based. The authors should compare against the MLE model baseline to demonstrate that in some situations their approach is better than the most naive MLE model-based approach.
* Proposition 3.2 uses this introduced notion of epsilon-strictly risk-sensitive. This definition is suitable for CVaR, but does not apply to many spectral risk measures, which apply non-zero mass to all quantiles (e.g. Wang risk measure). In the latter case, epsilon is zero, and therefore Proposition 3.2 does not support the arguments of the authors, as it shows the error is greater than or equal to zero. It seems like it should be possible to come up with a different bound for proposition 3.2 that is more general (and therefore shows that value-equivalence can be sub-optimal for any spectral risk measures other than expected value).

Minor comments:

* Reproducibility: the code provided has no Readme, and for the option trading environment it is completely unclear which of many files to run to generate the results in the paper.
* $\Pi$ is used to denote both a set of policies, and a projection operator. This notation is a little confusing.

Other than the lack of discussion about the potential advantages of the proposed approach (compared to MLE), the lack of a comparison to standard MLE baseline in the experiments, and uncertainty about some of the experiment details, I really like the paper. Thus, I am likely to increase my score to an accept score if I think these concerns are adequately addressed during the rebuttal period.

**Questions:**

* Are you able to provide experiments demonstrating the difference in performance between between risk-sensitive policies optimised using your distribution-equivalent models vs an MLE model? Perhaps we might expect that models learnt using your approach result in better risk-sensitive performance when model capacity is limited (as demonstrated in Grimm 2020 for the value-equivalence case).
* Can you explain the situations (such as limited model capacity, or limited data) where your approach is more suitable than MLE model learning for risk-sensitive policy learning?
* Section 5 does a good job of explaining that models can be equivalent only for a certain set of return statistics that are of interest. However, in the tabular experiments the authors learn an equivalent model for the mean and variance, and then optimise for CVaR. This seems to contradict section 5, as mean and variance are not sufficient for estimating CVaR. What was the motivation for choosing the mean and variance as the return statistics of interest in this experiment?
* What return statistics/functionals were used for the distribution-equivalent model in the option trading experiment?
* Is the improvement in sample efficiency (Line 329 of the paper) compared to the Lim & Malik (2022) approach because of the distributional-equivalence approach you have proposed, or simply because any model-based approach (such as MLE) is more efficient than Lim & Malik (2022)?
* How is the model represented in the experiments? In the tabular environment, I assumed it was just a categorical distribution over successor states. However, the appendix says that GPU training was used - indicating that this is a neural network model? Likewise, what was the model architecture for the option-training case?

**Limitations:**

I think the authors have done a good job of addressing potential limitations. In particular, by proposing the approximate version of distributional equivalence in Section 6.

---

> ### Author Rebuttal · Authors · 2023-08-10
>
> We thank the reviewer for their thorough review and very useful feedback, and are grateful for their positive comments on the paper. We address their concerns and questions below.
>
> ## Comparison to MLE
> We thank the reviewer for raising this, as we mainly focused our discussions on comparing to value equivalence, but we completely agree that a more focused comparison to MLE-based approaches will strengthen our paper. We added a discussion on this topic, which we provide below.
>
> The standard approach to learning a model is to use maximum likelihood estimation (MLE) based on data, which given a model class selects the model which is most likely to have produced the data seen. If the model class is expressive enough, and there is enough data, we may expect a model learnt using MLE to be useful for risk-sensitive planning. However, the success of this method relies on the model being able to model everything about the environment, which is an unrealistic assumption in general. In contrast, our method focuses on learning the aspects of the environment which are most relevant for risk-sensitive planning. With that in mind, we may expect our method to outperform the MLE when the model class is not expressive enough to model the entire environment, which may be due to a limited model class, or for example if the environment is very complex and impossible to model fully.
>
> ## Empirical comparison to MLE baselines
> We added an MLE baseline to all existing experiments (tabular and option trading) in the paper. We found that in these environments the MLE baseline performed approximately on par with our approach, and so we didn’t include these updated figures in the PDF for lack of space (although we will of course update the figures in the paper).
>
> To demonstrate empirically settings in which our approach out-performs naively using the MLE, we added a number of additional experiments:
> - We repeat the tabular experiments and constrain the model’s estimated transition matrix to be a certain rank, effectively restricting model capacity (Figure 2 in PDF) .
> - We repeat the option trading experiments, however we add additional dimensions to the state space which consist of uniform random noise, increasing the complexity of the environment to model (Figure 3 in PDF).
> - We repeat the option trading experiments, limiting the size of the hidden layer of the model, once again restricting model capacity (Figure 4 in PDF) .
>
> In each of these settings, our approach out-performed the MLE baseline, demonstrating our arguments from the previous point.
>
> ## Generalizing Proposition 3.2
> We thank the reviewer for raising this point, as we believe that their feedback has strengthened the impact of our result. The reviewer is correct that we formulated the proposition with CVaR in mind, and as such it is limited to the range of risk measures it is applicable to. We have generalized the proposition, and present it below, so that it is now applicable to all spectral risk measures.
>
> We say that a spectral risk measure $\varphi$ is $(\varepsilon, \delta)$-strictly risk sensitive if it corresponds to a function $\varphi$ such that $\varphi(\varepsilon) \leq \delta $. We note that our previous definition corresponds to the case that $\delta=0$. Moreover, this new definition is applicable to all spectral risk measures, in the sense that for any spectral risk measure there exists an $(\varepsilon, \delta)$ pair satisfying the definition. With this new definition, the bound $\frac{R_{max}}{1-\gamma} \varepsilon$ is replaced by $\frac{R_{max}}{1-\gamma} \varepsilon (1-\delta(1-\varepsilon))$. In particular, as the reviewer mentioned, this now provides a non-zero bound for all spectral risk measures other than expectation.
>
> ## Return statistics/risk functionals used in tabular experiments
> For the tabular experiments, we used the two moment functional due to the fact that it is Bellman-closed, so that it can be learnt exactly in a dynamic programming fashion, while there is no known Bellman-closed functional equivalent to CVaR, so learning it in a dynamic programming fashion will lead to a biased result. Of course, as pointed out by the reviewer, using the two moment functional to plan for CVaR may result in error due to the fact that the first two moments are not sufficient for estimating CVaR. We will make this more clear in the paper, and we have also expanded the tabular experiments to additionally learn a CVaR-equivalent model (although it is biased as discussed before), which we present in Figure 1 of PDF attached to the official comment.
>
> ## Return statistics used for option trading
> The statistical functional used in the option trading experiment is the functional $\psi=(F_{\mu}^{-1}(\tau_1), \dots, F_{\mu}^{-1}(\tau_m))$, where $\tau_i=(2i-1)/2m$, where $m=100$. In particular, our implementation of QR-DQN learns this functional $\psi$ of the return in order to take actions.
>
> We added a section in the appendix to discuss this in detail, and discuss that in a general sense our method can be combined with a model-free algorithm which learns a functional $\psi$ of the return to obtain a $\psi$-equivalent model.
>
> ## Experiment descriptions
> The model for tabular experiments was an exact categorical distribution over states, and the model for the option trading environment was a Gaussian transition model. We will explicitly describe these in the appendix, and modify Appendix E to provide CPU-hours for the tabular experiments rather than GPU-hours.
>
>
> ## Minor comments
> We agree that in its original state the code was not clear how to reproduce our results, we re-organized the code and included a readme with instructions to reproduce each figure.
> We apologize for the overloading of $\Pi$, we replaced the use of $\Pi$ for projection with $\operatorname{Proj}$.

---

> > ### Comment · Reviewer_B3wZ · 2023-08-10
> >
> > Well done on this strong rebuttal, and thank you for addressing all of the points that I raised in my review.
> >
> > I think the new results are great, and demonstrate that there is practical utility to this approach in additional to the theoretical contributions. I also appreciate the improvement to Proposition 3.2.
> >
> > In light of these improvements, I now believe that the paper should be accepted. I will update my score to a 7.

---

### Official Review · Reviewer_du98 · 2023-07-06

**Soundness:** 3 good
**Presentation:** 3 good
**Contribution:** 3 good
**Rating:** 5
**Confidence:** 3

**Summary:**

This paper studies the intersection of model-based RL and risk-sensitive RL. Firstly, the authors theoretically demonstrate that proper value equivalence can only plan optimally in the risk-neutral setting, and its performance will deteriorate as the risk level increases. Then the authors introduce distributional equivalence principle and prove that distributional equivalent models can be used for optimal planning with any risk measure. However, due to the inherent challenges of distribution, learning distributional equivalent models is not practical. Therefore, the authors combine the statistical functionals and propose statistical functional equivalence, which is parameterized by the choice of a statistical functional. The authors further demonstrate that the choice of a statistical functional determines the risk measures that can be used for optimal planning, and provide the loss functions for learning these models. Additionally, the authors show how the proposed framework can be integrated with existing model-free risk-sensitive algorithms. Finally, the authors validate the performance of the framework in both tabular experiments and option trading scenarios.

**Strengths:**

1. The paper is well written. The original contributions are highlighted clearly.
2. This paper demonstrates clear logic and presents a series of comprehensive theoretical proofs to the validity of the proposed methods.
3. The structure of this paper is complete, it provides an illustrative example that is simple and easy to understand.


**Weaknesses:**

1. The selection of parameters of the experimental environments and algorithms is not clearly given, such as the reward setting in Four rooms and the parameter settings of various methods in Option Trading.
2. The work in the experimental part is insufficient, and there is no more presentation of experimental results in the appendix, which causes the lack of persuasion.
3. I think the authors can add some enhanced verification experiments on the performance of more model-free risk-sensitive RL algorithms augmented with the proposed framework.


**Questions:**

1. What is the distinction between the spectral risk measures used in the paper and the distorted expectation risk measures?
2. Why is it necessary for the weight function of spectral risk measures to satisfy a non-increasing condition?
3. Tabular experiments only show the performance comparison with respect to ($\operatorname{CVaR}(0.5)$). I think it's better to show the expected returns as well.
4. For the combination of the modification of QR-DQN and statistical functional equivalent models, it is better to provide a description of the procedures of the algorithm for understanding.
5. When using statistical functional equivalent models to augment model-free risk-sensitive RL algorithms, will the data generated by the model be added to the replay buffer to improve sample efficiency during training?
6. In Line 110, $\mu$ in equation $F_\mu^{-1}(u)=\inf \{z \in \mathbb{R}: \mu(-\infty, z] \geq u\}$ seems to have a different meaning from $\operatorname{CVaR}_\tau(\mu)=\underset{Z \sim \mu}{\mathbb{E}}\left[Z \mid Z \leq F_\mu^{-1}(\tau)\right]$. Is this true?
7. The formulas in the paper lack proper numerical labels. Please improve it.
8. In Line 126, I think there may be something wrong when you are calculating the equation $\eta^{\pi^b}(x)=U([-2,2])$, because the superposition of uniform distributions should result in a triangular distribution. Can you show me your detailed calculation steps?

**Limitations:**

I think the authors need to add more discussion around the limitations of their approach. Specifically, can the proposed framework augment any model-free risk-sensitive RL algorithm? Will statistical functional equivalence limit the risk measures that can be used? A discussion on this point would be useful.

---

> ### Author Rebuttal · Authors · 2023-08-10
>
> We thank the reviewer for their time and effort in reviewing our paper, and we are grateful for your positive feedback on the paper's clarity, original contributions, and comprehensive theoretical proofs. We address the weaknesses and questions below.
>
> ## Environment selection and parameters
> We currently describe the environments in detail in Appendix E.1., if the reviewer believes we are missing details there we will happily add them. Regarding the hyperparameters of the option trading algorithm, we use the same hyperparameters as was used in [1]. We will specify this in the appendix.
>
> ## Insufficient empirical results
> Thank you for your comment, we have taken the following steps to enhance our empirical support:
> - We added experiments highlighting the benefits of our method over learning models using MLE:
> - We repeat the tabular experiments and constrain the model’s estimated transition matrix to be a certain rank, effectively restricting model capacity (Figure 2 in PDF) .
> - We repeat the option trading experiments, however we add additional dimensions to the state space which consist of uniform random noise, increasing the complexity of the environment to model (Figure 3 in PDF).
> - We repeat the option trading experiments, limiting the size of the hidden layer of the model, once again restricting model capacity (Figure 4 in PDF) .
> - If there are any other experiments that the reviewer believes should be included, we are open to consider it.
>
> ## Comparison to more model-free baselines
> We chose to use the algorithm from [1] as a baseline to illustrate how our framework can be combined with an existing model-free algorithm. If the reviewer has a risk-sensitive model-free algorithm in mind which they believe would be illustrative to include as an additional baseline, we would be happy to include it.
>
> ## Additional limitations
> We thank the reviewer for pointing these out, and we added discussion to both points raised. Regarding “*Specifically, can the proposed framework augment any model-free risk-sensitive RL algorithm?*”, we added a section in the appendix discussing this point, and demonstrating how our framework can augment a model-free algorithm which learns a statistical functional $\psi$ of the return with a $\psi$-equivalent model. Regarding “*Will statistical functional equivalence limit the risk measures that can be used?*”, we previously touched on this in Appendix F, but we have since expanded the discussion. In particular, our theory demonstrates that statistical functional equivalence can be used for any risk measure which is in the span of the statistical functional used (in the sense of Proposition 5.10.). However, our experiments demonstrate that in practice, statistical functionals can often plan near-optimally for risk measures not in their span, for example the moments functional planning for CVaR in the tabular domain. We highlighted understanding approximate planning in this sense as a direction for future work.
>
>
> ## Questions
> 1. Spectral risk measures and distorted expectation risk measures are related formulations, as they both weighted integrals of the quantile function $F^{-1}_{\mu}$. Spectral risk measures are a proper subset of distortion risk measures, as shown in [2], this is reflective of the fact that spectral risk measures are all coherent (see the following answer for a discussion of this term), while distorted expectation risk measures are not.
> 2. The requirement that the weighting function $\varphi$ is non-increasing is required so that spectral risk measures are coherent [3]. A coherent risk measure is intuitively one that satisfies a collection of properties which makes it a ‘desirable’ measure for decision making.
> 3. We agree with this suggestion, and we have added figures with the expected returns as well (Figure 1 in PDF).
> 4. To increase the clarity of our architecture combined with QR-DQN, we will add a section with a detailed description in the appendix, as described in the previous section (additional limitations).
> 5. In our experiments, we sample from the replay buffer, and replace the real next states with our model’s predicted next states, the same method which was used in [4]. We remark that this is a rather simple way to use the model, and future work can be done to find more sophisticated techniques of using the model. We will make this clear in the paper, and highlight it as a potential for future work.
> 6. In both equations, $\mu$ is the same real probability measure. We use $\mu(A)$ to indicate the measure of a set $A$, and $\mathbb{E}_{X\sim \mu}[f(X)]$ to indicate the expected value of $f$ under $\mu$. We will make this more clear in the paper.
> 7. We will add numerical labels to equations for increased clarity.
> 8. The calculation that $\eta^{\pi^b}(x)=U([-2,2])$ is presented as Example 2.10. of [5], we refer the reviewer to this reference as we believe we would not be able to reproduce it as clearly as it is shown there. We note that one possible source of confusion is that a triangular distribution would result from the sum of continuous uniform random variables $U([-1, 1])$, while the quantities being added here are discrete uniform random variables $U(\\\{-1, 1\\\})$.
>
>
> [1] Lim, S. H. and Malik, I. Distributional reinforcement learning for risk-sensitive policies. Advances in Neural Information Processing Systems, 2022.
>
> [2] Gzyland, H. and Mayoral, S. On a relationship between spectral and distorted risk measures.
> Spanish Finance Association, 2016.
>
> [3] Artzner, P., Delbaen, F., Jean-Marc, E., and Heath, D. Coherent measures of risk. Mathematical Finance, 1999.
>
> [4] Grimm, C., Barreto, A., Singh, S., and Silver, D. The value equivalence principle for model-based reinforcement learning. Advances in Neural Information Processing Systems, 2020.
>
> [5] Bellemare, M. G., Dabney, W., and Rowland, M. Distributional Reinforcement Learning. MIT Press, 2023. http://www.distributional-rl.org.

---

> > ### Comment · Reviewer_du98 · 2023-08-15
> >
> > I would like to thank the authors for providing responses to my questions.
> >
> > I still believe that the experimental design and results in the paper are insufficient, causing its limited persuasiveness.
> >
> > Therefore, I will maintain my initial rating.

---

> > > ### Author Response · Authors · 2023-08-18
> > > **Thanks! Concrete suggestion for experiments?**
> > >
> > > We appreciate your response. Thank you!
> > >
> > > Since you mentioned that the experimental design and results in the paper are insufficient, even after our new results during the rebuttal phase, we would like to kindly ask you if you have any specific experiments in mind that you would like to see? We will consider your suggestions in our future revisions, so that we have a more convincing paper.

---

### Official Review · Reviewer_us8H · 2023-07-06

**Soundness:** 3 good
**Presentation:** 3 good
**Contribution:** 3 good
**Rating:** 7
**Confidence:** 4

**Summary:**

The authors propose to extend the notion of value equivalence to distributional model equivalence for the purpose of risk-sensitive reinforcement learning. Theoretically, the paper first shows that value equivalence is insufficient for planning risk-sensitive policies, then introduces both exact and approximate versions of distributional model equivalence, along with some of their theoretical properties. Empirically, the utility of the proposed approach is demonstrated on a number of simple domains for evaluating risk-sensitive policies.

**Strengths:**

The main theoretical contribution of the paper is in generalizing the notion of value equivalence to distributional model equivalence for the purpose of learning risk-sensitive policies. This is well-motivated and the presentation is clear.

**Weaknesses:**

The empirical evaluations are rather limited. It seems that sample efficiency is the key motivation for the entire approach, yet this is not clearly demonstrated by the examples in section 7. It might be more illuminating if one could see how the performance changes with respect to the number of training samples (actual and/or modeled).

**Questions:**

What about policy gradient approaches? Is the proposed notion of distributional model equivalence still relevant?

---

> ### Author Rebuttal · Authors · 2023-08-10
>
> We thank the reviewer for their useful comments, and are pleased to hear that they found our paper well-motivated and clearly presented. We address the highlighted weakness and questions below.
>
> ## Empirical evaluations
> To address the reviewer’s concerns regarding limited evaluation, we made a number of modifications:
> - We added sample efficiency curves for the option trading experiments to the appendix (not included in PDF of figures due to lack of space).
> - We made the discussion more clear so that the main motivation is not only sample efficiency, but also the benefits of our framework over a naive MLE model. We demonstrated this in Figures 2, 3, and 4 of the PDF attached to the top-level comment.
>
> ## Applications to policy gradient
> While we focused on the value-based setting in this paper, we believe that distributional model equivalence may be adapted for the policy optimization case, so that it may be useful for risk-sensitive policy gradient formulations such as [1]. Our current work would likely need to be adapted in a similar fashion as the construction in [2]. We will highlight this as a potential direction for future work. In the actor-critic setting, our current method can be used to learn a better critic, which can then contribute to learning an improved actor. This can be especially useful for  a risk-sensitive actor critic framework such as [3].
>
> [1] Aviv Tamar, Yinlam Chow, Mohammad Ghavamzadeh, Shie Mannor. Policy Gradient for Coherent Risk Measures, NeurIPS, 2015.
>
> [2] Romina Abachi, Mohammad Ghavamzadeh, and Amir-massoud Farahmand. Policy-aware model learning for policy gradient methods, arXiv, 2020.
>
> [3] Prashanth L.A., Mohammad Ghavamzadeh, Actor-Critic Algorithms for Risk-Sensitive MDPs, NeurIPS, 2013.

---

> > ### Comment · Reviewer_us8H · 2023-08-18
> >
> > I thank the authors for the rebuttal. I'll keep my score.

---

### Author Rebuttal · Authors · 2023-08-10

We thank all of the reviewers for their time and effort spent reviewing and the feedback provided. We believe that based on their feedback, we were able to significantly improve the quality of our work.

We now highlight some of the main modifications made.
- As suggested by Reviewer B3wZ21, we added discussion on the benefits of our method over model learning using an MLE baseline, and which settings one would see a benefit. We further added a number of experiments to corroborate our reasoning which can be found in Figures 2, 3, and 4 of the attached PDF.
- As suggested by Reviewer B3wZ21, we generalized Proposition 3.2. so that it is applicable to any spectral risk measure, while previously it only applied to CVaR-like risk measures. In particular, the bound now provides a non-zero optimality gap for any spectral risk measure other than expectation.
- As suggested by Reviewers du98 and B3wZ21, we added a section in the appendix clarifying the experimental setup of how QR-DQN was combined with our model-learning framework, and more generally how a model-free algorithm which learns a statistical functional $\psi$ of the return can be augmented with our method to learn a $\psi$-equivalent model.

---

### Decision · Program_Chairs · 2023-09-21

**Decision:**

Accept (poster)

**Comment:**

The review scores are 7, 7, 5, with an average score of 6.33.

This paper proposes an interesting approach for risk sensitive reinforcement learning through a distribution equivalence principle. It first presents the limitation of value equivalence for risk sensitive planning, then introduces the distribution equivalence principle, and then proposes a tractable way of using this idea for risk sensitive RL.

I have read this paper, reviewer comments, and the authors’ rebuttal carefully.  The paper is technically solid. The proofs are fairly involved and not straight forward. The main idea, risk sensitive RL through distributional equivalence, is novel, and will be of interest to the sub-area of robust/risk RL. All the reviewers are in agreement about the solid technical contributions of this paper.

The main concerns, expressed by Reviewer us8H and Reviewer du98, are about the lack of sufficient empirical evaluations. In response to the reviewers’ comments, the authors have added multiple additional experiments in their rebuttal. They have also mentioned the changes they will make in their revision, including more details about the experiment settings. From my reading, these additional experiments seem to have addressed most of the questions by the reviewers. The reviewers are also satisfied with rebuttal.

For the final version, please make all the changes you have promised and update the paper.

Regards,

AC